# Learning a distance measure from the information-estimation geometry of data

**Guy Ohayon**
Flatiron Institute
gohayon@flatironinstitute.org

**Pierre-Etienne H. Fiquet**
Flatiron Institute
pfiquet@flatironinstitute.org

**Florentin Guth**
Flatiron Institute
New York University
florentin.guth@nyu.edu

**Jona Ballé**
New York University
jona.balle@nyu.edu

**Eero P. Simoncelli**
Flatiron Institute
New York University
eero.simoncelli@nyu.edu

## Abstract

We introduce the Information-Estimation Metric (IEM), a novel form of distance function derived from an underlying continuous probability density over a domain of signals. The IEM is rooted in a fundamental relationship between information theory and estimation theory, which links the log-probability of a signal with the errors of an optimal denoiser, applied to noisy observations of the signal. In particular, the IEM between a pair of signals is obtained by comparing their denoising error vectors over a range of noise amplitudes. Geometrically, this amounts to comparing the score vector fields of the *blurred* density around the signals over a range of blur levels. We prove that the IEM is a valid global distance metric and derive a closed-form expression for its local second-order approximation, which yields a Riemannian metric. For Gaussian-distributed signals, the IEM coincides with the Mahalanobis distance. But for more complex distributions, it adapts, both locally and globally, to the geometry of the distribution. In practice, the IEM can be computed using a learned denoiser (analogous to generative diffusion models) and solving a one-dimensional integral. To demonstrate the value of our framework, we learn an IEM on the ImageNet database. Experiments show that this IEM is competitive with or outperforms state-of-the-art supervised image quality metrics in predicting human perceptual judgments.

## 1 Introduction

Distance functions are central to many scientific and engineering enterprises, enabling systematic comparison, organization, and interpretation of data. In some cases, a meaningful notion of distance arises from the structure or distribution of the data (*e.g.*, geodesics in Riemannian manifolds, z-scores for Gaussian distributions), or from the requirements of the task (*e.g.*, Hamming distance in error-correcting codes, edit distance in text processing). However, this is often not the case. For instance, algorithms that process natural signals (*e.g.*, compression engines) should ideally be evaluated in terms of human perception, for which no precise mathematical definition is available. Numerous algorithms aiming to mimic human perception have been proposed (Wang et al., 2004; Heusel et al., 2017; Zhang et al., 2018; Ding et al., 2022; Chen et al., 2024), with the most successful approaches to date being those trained (supervised) on databases of human perceptual judgments. Nevertheless, this reliance on human-labeled data is problematic, as data annotation is a highly costly and noisy procedure. More importantly, supervised approaches are difficult to interpret mathematically, making it harder to explain the principles that underlie our perceptual judgments of similarity between natural signals (Barlow, 1989). Deriving a perceptual metric solely based on unlabeled data remains a fundamental open problem of both scientific and practical importance.

A natural opportunity for developing such a metric arises from the concept of coding efficiency. Biological sensory systems are believed to decompose incoming signals in a manner that maximizes the transmission of information about those signals, subject to biological constraints (*e.g.*,

noise, metabolic cost) (Attneave, 1954; Barlow, 1961; Laughlin, 1981; van Hateren, 1992; Atick & Redlich, 1992; Olshausen & Field, 1996; Barlow, 2001; Simoncelli & Olshausen, 2001). Put differently, sensory pathways function as communication channels optimized for natural signals, implying that our ability to discriminate between natural signals depends on their statistical properties. Indeed, for one-dimensional sensory attributes, previous work has shown that perceptual sensitivity to small signal perturbations increases with the probability of the signal (Laughlin, 1981; Ganguli & Simoncelli, 2014; Wei & Stocker, 2017). However, in the multivariate setting, such as color discrimination (*i.e.*, detecting changes in hue or saturation), humans exhibit complex patterns of sensitivity that vary with the *direction* of the signal's perturbation (MacAdam, 1942). This leaves us with a conundrum: how can a probability density, which is a scalar function, induce a Riemannian metric, let alone a global distance function between any pair of signals in the domain?

In an attempt to construct a distance function from a probability density, it is natural to resort to principles from information theory. Unfortunately, information-theoretic quantities are agnostic to the *geometry* of the probability distribution. For example, the mutual information between random variables is invariant to bijective (even discontinuous) transformations of the variables. In contrast, estimation quantities such as denoising error are explicitly tied to the geometry of the density through an assumed observation model (*e.g.*, additive Gaussian noise) and loss function (*e.g.*, square error). Despite this salient difference, a line of work (Guo et al., 2005; 2013) rooted in information theory (Stam, 1959) and empirical Bayesian methods (Robbins, 1956) has revealed an extensive correspondence between these seemingly unrelated quantities. It takes the form of a set of relationships that express information quantities in terms of estimation quantities, thereby linking probability (information) with geometry (the shape of the "data support"). In particular, *scalar* probability values can be decomposed into denoising error *vectors*, which provide a natural way to characterize the geometry of the signal density. Indeed, denoising errors are proportional to the *score* (gradient of the log) of the signal density *blurred* through convolution with a Gaussian density. This relationship between denoising errors and scores, known as the Tweedie–Miyasawa formula (Robbins, 1956; Miyasawa, 1961), is the foundation of generative diffusion models (Sohl-Dickstein et al., 2015; Ho et al., 2020; Song et al., 2020).

Building on these information–estimation relationships, we introduce a novel form of distance function which is derived from the geometry of a given probability density. Our distance, coined the Information-Estimation Metric (IEM), compares the score vector fields of the blurred density in the vicinity of two given signals. More specifically, it is defined as the mean square error (MSE) between these score vector fields, integrated over a range of blur levels (*i.e.*, Gaussian noise magnitudes). We prove that the IEM is a valid distance metric (in the mathematical sense), and show that it coincides with the Mahalanobis distance (Mahalanobis, 1936) when the prior density is Gaussian. For more complex priors, however, the IEM reflects the structure of the "data support"—adapting to the *global geometry* of the density. Furthermore, we analyze the local behavior of the IEM by deriving the second-order expansion of the distance between a signal and its perturbed version, which yields a Riemannian metric. We show that this Riemannian metric is most sensitive (1) in regions where the curvature of the log-density is highest, and (2) to perturbations that induce the largest changes in the signal's probability. This implies that the IEM behaves like a *locally adaptive* Mahalanobis distance—conforming to the *local geometry* of the density. Importantly, the IEM can be efficiently learned from samples by training a denoiser (*i.e.*, a diffusion model). We train such a denoiser on ImageNet (Deng et al., 2009) and use it to compute the IEM. Although the IEM is learned unsupervised from unlabeled image data, we find that it is competitive with supervised perceptual distance measures in terms of predicting human judgments of image similarity.

## 2 THE INFORMATION-ESTIMATION METRIC

We aim to construct a distance function that is induced by the geometry of a given probability density. In information theory, it is natural to compare two signals $x_1$ and $x_2$ using their log-probability ratio, which may be turned into a "distance" by taking its square value. However, this is a poor choice, as it depends solely on the (scalar) values of the density at the two points. Instead, we would like a distance measure that is associated with the *geometry* of the density (*e.g.*, the curvature of the density around the two signals). To this end, we build upon a fundamental equation that *decomposes* the log-probability of a signal in terms of the geometry of the probability density in the

vicinity of the signal. We then apply this decomposition to the log-probability *ratio* of two signals, yielding a distance metric that adapts to the density's geometry.

**Observation channel.**   Let $p_{\mathbf{x}}$ denote the probability density function of a random vector $\mathbf{x}$ taking values in $\mathbb{R}^d$ (*i.e.*, the signal). To decompose $p_{\mathbf{x}}$, we introduce an observation process $\mathbf{y}_\gamma$ such that $p_{\mathbf{y}_\gamma}$ gradually "zooms" into $p_{\mathbf{x}}$ as the signal-to-noise ratio (SNR) $\gamma$ is increased, analogously to how diffusion models generate samples. Specifically, we define $\mathbf{y}_\gamma$ as a Gaussian channel

$$\mathbf{y}_\gamma = \gamma\mathbf{x} + \mathbf{w}_\gamma, \tag{1}$$

where $\mathbf{w}_\gamma \sim \mathcal{N}(\mathbf{0}, \gamma\boldsymbol{I})$ is a standard Wiener process which is independent of $\mathbf{x}$. Since the noise $\mathbf{w}_\gamma$ is statistically independent of $\mathbf{x}$, the distribution $p_{\mathbf{y}_\gamma}$ is obtained by *blurring* $p_{\mathbf{x}}$ through convolution with the Gaussian density $p_{\mathbf{w}_\gamma}$. Viewing $\log p_{\mathbf{y}_\gamma}(\mathbf{y}_\gamma)$ as a stochastic process that evolves with $\gamma$, we can decompose $\log p_{\mathbf{x}}(\mathbf{x})$ in terms of the *increments* of this process. By combining two fundamental relations from previous work, we show next that these increments characterize the geometry of $\log p_{\mathbf{x}}$ in the vicinity of $\mathbf{x}$.

**Pointwise I-MMSE.**   Venkat & Weissman (2012) proved that $\log p_{\mathbf{y}_\Gamma}(\mathbf{y}_\Gamma)$ for any fixed $\Gamma > 0$ can be expressed in terms of the denoising error vectors of the minimum mean square error (MMSE) estimator of $\mathbf{x}$ from $\mathbf{y}_\gamma$, $\mathbb{E}[\mathbf{x} \,|\, \mathbf{y}_\gamma]$, integrated across all SNR levels $\gamma \in [0, \Gamma]$. Formally,

$$-\log p_{\mathbf{y}_\Gamma}(\mathbf{y}_\Gamma) = \int_0^\Gamma (\mathbf{x} - \mathbb{E}[\mathbf{x}\,|\,\mathbf{y}_\gamma]) \cdot \mathrm{d}\mathbf{w}_\gamma + \frac{1}{2}\int_0^\Gamma \|\mathbf{x} - \mathbb{E}[\mathbf{x}\,|\,\mathbf{y}_\gamma]\|^2 \mathrm{d}\gamma - \log p_{\mathbf{w}_\Gamma}(\mathbf{w}_\Gamma), \tag{2}$$

where this equality holds with probability one (almost surely), *i.e.*, it holds *pointwise* for almost every realization of the signal $\mathbf{x} = \boldsymbol{x}$ and of the Wiener process trajectory $\{\mathbf{w}_\gamma = \boldsymbol{w}_\gamma\}_{\gamma=0}^\Gamma$. Equation (2), which we refer to as the pointwise I-MMSE formula, is a generalization of the I-MMSE formula (Guo et al., 2005), whose roots date back to de Bruijn's identity from the 1950s (Stam, 1959, see App. A.2 for more detailed background). When $\Gamma \to \infty$, Eq. (2) expresses the log-density of the original signal, $\log p_{\mathbf{x}}(\mathbf{x})$, in terms of the denoising errors at all $\gamma \in [0, \infty)$.

**Geometric interpretation.**   Denoising errors are related to the *gradients* of $\log p_{\mathbf{y}_\gamma}$, *i.e.*, the scores of the blurred density $p_{\mathbf{y}_\gamma}$, via the Tweedie–Miyasawa formula (Robbins, 1956; Miyasawa, 1961):

$$\mathbf{x} - \mathbb{E}[\mathbf{x}\,|\,\mathbf{y}_\gamma] = -\frac{1}{\gamma}\mathbf{w}_\gamma - \nabla \log p_{\mathbf{y}_\gamma}(\mathbf{y}_\gamma), \tag{3}$$

where the gradient on the right-hand side is taken w.r.t. $\mathbf{y}_\gamma$. Substituting this formula into Eq. (2), we now see that $\log p_{\mathbf{x}}(\mathbf{x})$ (a *scalar*) can be decomposed in terms of the local geometry of $\log p_{\mathbf{y}_\gamma}(\mathbf{y}_\gamma)$, particularly the *gradients* $\nabla \log p_{\mathbf{y}_\gamma}(\mathbf{y}_\gamma)$, at all SNR levels $\gamma$. We refer to this decomposition as the *information-estimation geometry* of the density $p_{\mathbf{x}}$.

**Definition of the Information-Estimation Metric (IEM).**   The relationships above suggest a natural way to compare two arbitrary points $\boldsymbol{x}_1$ and $\boldsymbol{x}_2$, by tracking the increments of their log-probability ratio under the blurred density, $\log\left(p_{\mathbf{y}_\gamma}(\gamma\boldsymbol{x}_1 + \mathbf{w}_\gamma)/p_{\mathbf{y}_\gamma}(\gamma\boldsymbol{x}_2 + \mathbf{w}_\gamma)\right)$. Doing so amounts to comparing the local geometry of $\log p_{\mathbf{x}}$ around $\boldsymbol{x}_1$ and $\boldsymbol{x}_2$, as illustrated in Fig. 1. Specifically, by combining Eqs. (2) and (3), we obtain

$$\log\left(\frac{p_{\mathbf{y}_\Gamma}(\Gamma\boldsymbol{x}_1 + \mathbf{w}_\Gamma)}{p_{\mathbf{y}_\Gamma}(\Gamma\boldsymbol{x}_2 + \mathbf{w}_\Gamma)}\right) = \int_0^\Gamma \left(\nabla \log p_{\mathbf{y}_\gamma}(\gamma\boldsymbol{x}_1 + \mathbf{w}_\gamma) - \nabla \log p_{\mathbf{y}_\gamma}(\gamma\boldsymbol{x}_2 + \mathbf{w}_\gamma)\right) \cdot \mathrm{d}\mathbf{w}_\gamma$$
$$- \frac{1}{2}\int_0^\Gamma \left(\left\|\nabla \log p_{\mathbf{y}_\gamma}(\gamma\boldsymbol{x}_1 + \mathbf{w}_\gamma) + \mathbf{w}_\gamma/\gamma\right\|^2 - \left\|\nabla \log p_{\mathbf{y}_\gamma}(\gamma\boldsymbol{x}_2 + \mathbf{w}_\gamma) + \mathbf{w}_\gamma/\gamma\right\|^2\right)\mathrm{d}\gamma. \tag{4}$$

Since Eq. (4) is an Itô process, it is natural to quantify the sum of its squared increments by taking the expected *quadratic variation* of the process, which is simply the second moment of the diffusion coefficient integrated over the range $\gamma \in [0, \Gamma]$. This leads to our proposed distance function.

**Definition 1.** *The* Information-Estimation Metric (IEM) *induced by the density $p_{\mathbf{x}}$ is defined as*

$$\mathrm{IEM}(\boldsymbol{x}_1, \boldsymbol{x}_2, \Gamma) := \left(\int_0^\Gamma \mathbb{E}\left[\left\|\nabla \log p_{\mathbf{y}_\gamma}(\gamma\boldsymbol{x}_1 + \mathbf{w}_\gamma) - \nabla \log p_{\mathbf{y}_\gamma}(\gamma\boldsymbol{x}_2 + \mathbf{w}_\gamma)\right\|^2\right]\mathrm{d}\gamma\right)^{\frac{1}{2}}$$

*where the expectations are taken over $p_{\mathbf{w}_\gamma}$ for each $\gamma$.*

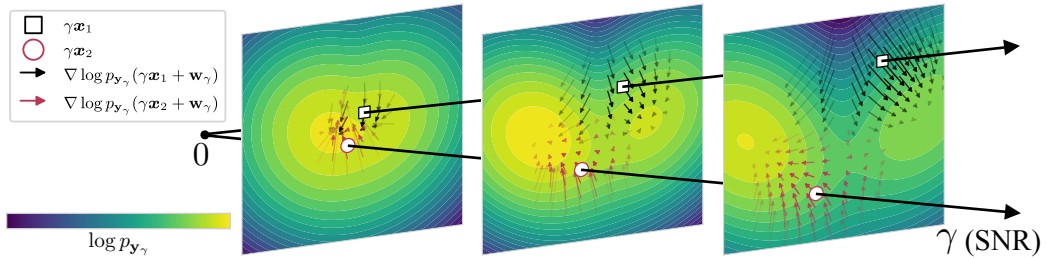

Figure 1: **The information-estimation geometry around two points.** We show a Gaussian mixture log-density and its gradient vector fields around the points $\gamma\boldsymbol{x}_1$ and $\gamma\boldsymbol{x}_2$ for three different SNR levels $\gamma$. The space is rescaled by $\gamma$ and the distribution collapses to a point at $\gamma = 0$. When blurring the density (small $\gamma$), the two modes merge, and the gradients around $\gamma\boldsymbol{x}_1$ point toward either of the modes. When the two modes are far enough apart (large $\gamma$), most gradient vectors point toward their closest mode. Thus, the local gradients around a given point can capture different geometrical features of the distribution, depending on the SNR $\gamma$. The Information-Estimation Metric (IEM, Def. 1) between the two points $\boldsymbol{x}_1$ and $\boldsymbol{x}_2$ is the square error between the local gradient fields around them, weighted by a Gaussian window (illustrated by the opacity of the gradients' arrows) and integrated over all levels of SNR $\gamma \in [0, \Gamma]$.

For ease of notation, we write $\mathrm{IEM}(\boldsymbol{x}_1, \boldsymbol{x}_2) \coloneqq \mathrm{IEM}(\boldsymbol{x}_1, \boldsymbol{x}_2, \infty)$. Although our construction does not make it obvious, the IEM is a proper distance metric (see proof in App. C.1):

**Theorem 1.** *For every $\Gamma > 0$, the IEM is a proper distance metric: it is symmetric, non-negative, equal to zero if and only if $\boldsymbol{x}_1 = \boldsymbol{x}_2$, and it satisfies the triangle inequality.*

In App. B, we discuss an intriguing relationship between the IEM and the Kullback–Leibler (KL) divergence between distributions. Specifically, we interpret the IEM as a local decomposition of the KL divergence between two translated copies of $p_{\mathbf{x}}$, centered at $\boldsymbol{x}_1$ and $\boldsymbol{x}_2$. We also define a *mismatched* IEM, generalizing the IEM to the case where $\boldsymbol{x}_1$ and $\boldsymbol{x}_2$ are assumed to come from *different* distributions.

## 2.1 LOCAL GEOMETRY

To gain insight into the properties of the IEM, we study its local behavior, namely the distance between a given signal $\boldsymbol{x}$ and its perturbation $\boldsymbol{x} + \boldsymbol{\epsilon}$ for small $\boldsymbol{\epsilon}$. As for any distance, $\boldsymbol{\epsilon} = 0$ is a global minimum, and we can express the quadratic expansion of the IEM in $\boldsymbol{\epsilon}$ as $\mathrm{IEM}^2(\boldsymbol{x}, \boldsymbol{x} + \boldsymbol{\epsilon}, \Gamma) = \boldsymbol{\epsilon}^\top \boldsymbol{G}(\boldsymbol{x}, \Gamma)\boldsymbol{\epsilon} + o(\|\boldsymbol{\epsilon}\|^2)$. The positive-definite matrix $\boldsymbol{G}(\boldsymbol{x}, \Gamma)$ then acts as a local metric (in the Riemannian sense), but note that its relationship with the IEM is one-way: the IEM is *not* equivalent to the geodesic distance that corresponds to $\boldsymbol{G}(\boldsymbol{x}, \Gamma)$. The local metric $\boldsymbol{G}(\boldsymbol{x}, \Gamma)$ is characterized in the following theorem (see proof in App. C.2):

**Theorem 2.** *The local Riemannian metric derived from the second-order Taylor expansion of the squared* IEM *is given by*

$$\boldsymbol{G}(\boldsymbol{x}, \Gamma) = \int_0^\Gamma \gamma^2 \mathbb{E}\left[\left(\nabla^2 \log p_{\mathbf{y}_\gamma}(\gamma\boldsymbol{x} + \mathbf{w}_\gamma)\right)^2\right] \mathrm{d}\gamma \tag{5}$$

$$= \int_0^\Gamma \mathbb{E}\left[\left(\boldsymbol{I} - \gamma\mathrm{Cov}[\mathbf{x} \mid \mathbf{y}_\gamma = \gamma\boldsymbol{x} + \mathbf{w}_\gamma]\right)^2\right] \mathrm{d}\gamma, \tag{6}$$

*where the expectations are taken over $p_{\mathbf{w}_\gamma}$ for each $\gamma$. Moreover, for $\Gamma = \infty$ we have*

$$\mathbb{E}[\boldsymbol{G}(\mathbf{x})] = \mathbb{E}\left[-\nabla^2 \log p_{\mathbf{x}}(\mathbf{x})\right] = \mathbb{E}\left[\nabla \log p_{\mathbf{x}}(\mathbf{x})\nabla \log p_{\mathbf{x}}(\mathbf{x})^\top\right], \tag{7}$$

*where we denote $\boldsymbol{G}(\boldsymbol{x}) \coloneqq \boldsymbol{G}(\boldsymbol{x}, \infty)$ and the expectations are taken over $p_{\mathbf{x}}$.*

Theorem 2 gives two equivalent expressions for the local metric induced by the IEM. In particular, Eq. (5) shows how this local metric, $\boldsymbol{G}(\boldsymbol{x}, \Gamma)$, is tied to the local curvature of $\log p_{\mathbf{x}}$ around the point $\boldsymbol{x}$. Indeed, the Hessian $\nabla^2 \log p_{\mathbf{y}_\gamma}(\gamma\boldsymbol{x} + \mathbf{w}_\gamma)$, which is a (nonlinear) smoothing of

$\nabla^2 \log p_{\mathbf{x}}(\boldsymbol{x})$, describes the local curvature at blur level $\gamma$. In fact, the relationship between $\boldsymbol{G}(\boldsymbol{x}, \Gamma)$ and $\nabla^2 \log p_{\mathbf{x}}(\boldsymbol{x})$ becomes clearer when taking $\Gamma \to \infty$ and averaging over $p_{\mathbf{x}}$, as expressed by Eq. (7). Qualitatively, this demonstrates that, locally around the signal $\boldsymbol{x}$, the IEM is more sensitive to perturbations $\boldsymbol{\epsilon}$ that change the log-probability of $\boldsymbol{x}$ the most. Note that it is not true in general that $\boldsymbol{G}(\boldsymbol{x}) = -\nabla^2 \log p_{\mathbf{x}}(\boldsymbol{x})$ pointwise, as the Hessian of the log-density may not be negative semi-definite, whereas $\boldsymbol{G}(\boldsymbol{x}) \succeq 0$ by construction. The local metric $\boldsymbol{G}(\boldsymbol{x})$ thus acts as a positive semi-definite smoothing of $-\nabla^2 \log p_{\mathbf{x}}(\boldsymbol{x})$.

Furthermore, Eq. (6) relates the local metric $\boldsymbol{G}(\boldsymbol{x}, \Gamma)$ to the covariance of $\mathbf{x}|\mathbf{y}_\gamma = \gamma \boldsymbol{x} + \mathbf{w}_\gamma$, which is compared to $\boldsymbol{I}/\gamma$—the covariance of rescaled observation noise: $\boldsymbol{x} + \mathbf{w}_\gamma/\gamma \sim \mathcal{N}(\boldsymbol{x}, \boldsymbol{I}/\gamma)$. Equation (6) therefore provides additional intuition about the behavior of $\boldsymbol{G}(\boldsymbol{x}, \Gamma)$. First, when the noisy observations $\gamma \boldsymbol{x} + \mathbf{w}_\gamma$ can be effectively denoised across many SNR levels $\gamma$, the posterior covariance for such values of $\gamma$ is substantially smaller than that of the noise, which results in (relatively) high sensitivity to small perturbations of $\boldsymbol{x}$. A simple practical example of this scenario is when $\boldsymbol{x}$ is a "smooth" signal (*e.g.*, an image of a clear blue sky). Second, perturbations $\boldsymbol{\epsilon}$ that can be effectively denoised also lead to large local distance values. For instance, if the density $p_{\mathbf{x}}$ is supported on a low-dimensional manifold, then the local metric $\boldsymbol{G}(\boldsymbol{x}, \Gamma)$ is more sensitive around points $\boldsymbol{x}$ that are near the manifold, and in directions $\boldsymbol{\epsilon}$ that are orthogonal to the local tangent subspace.

## 2.2 ILLUSTRATIVE EXAMPLES

**Gaussian prior.** The IEM depends on the distribution of the data $p_{\mathbf{x}}$. When this distribution is Gaussian, $p_{\mathbf{x}} = \mathcal{N}(\boldsymbol{\mu}, \boldsymbol{\Sigma})$, and $\Gamma = \infty$, the IEM coincides with the well-known Mahalanobis distance (see App. C.3 for proof):

$$\text{IEM}(\boldsymbol{x}_1, \boldsymbol{x}_2) = \sqrt{(\boldsymbol{x}_1 - \boldsymbol{x}_2)^\top \boldsymbol{\Sigma}^{-1} (\boldsymbol{x}_1 - \boldsymbol{x}_2)}. \tag{8}$$

In other words, the IEM is the Euclidean distance after whitening the data: $\boldsymbol{x} \mapsto \boldsymbol{\Sigma}^{-\frac{1}{2}}(\boldsymbol{x} - \boldsymbol{\mu})$. Displacements in directions of small variance of the data are thus amplified and contribute more to the final distance, as visualized in the center column of Fig. 2.

This closed-form expression of the Gaussian IEM comes from the linearity of the corresponding optimal denoisers. While more complicated distributions $p_{\mathbf{x}}$ have non-linear optimal denoisers, they are often *locally* linear (Milanfar, 2013; Mohan et al., 2020), so that the corresponding IEM behaves like a Mahalanobis distance *locally*, adapting to the "local covariance" of the data. This is in agreement with our observations above about the local behavior of the IEM for general priors. Together, they paint a picture of how the IEM adapts to the geometry of the data distribution.

Since the IEM coincides with the Mahalanobis distance when $p_{\mathbf{x}}$ is Gaussian, it is important to examine the behavior of the IEM when $p_{\mathbf{x}}$ is no longer Gaussian.

**Gaussian mixture prior.** First, consider a two-dimensional, two-mode Gaussian mixture model. To compute the IEM and the local metric $\boldsymbol{G}(\boldsymbol{x}, \Gamma)$, we numerically solve the integrals in Def. 1 and Eq. (5), using closed-form expressions for $\log p_{\mathbf{y}_\gamma}$ and related quantities (see details in App. E.4). To illustrate the global behavior of the IEM, we choose a reference point $\boldsymbol{x}_{\text{ref.}}$, and evaluate its distance from each of a uniform grid of points $\boldsymbol{x}$. We then plot the resulting equidistant contours (Fig. 2, top row), and compare with a unimodal Gaussian density to illustrate how the IEM adapts to the prior. The IEM clearly adapts to the density's global geometry: the equidistant contours resemble the shape of the log-density contours. Interestingly, the regions delimited by equidistant contours can be disconnected: points belonging to one mode are closer to points belonging to the other mode than they are to points lying in between the modes. This is because the local curvature of the log-density can be similar in the vicinity of two points, even if their Euclidean distance is large.

Furthermore, we eigendecompose $\boldsymbol{G}(\boldsymbol{x}, \Gamma)^{-\frac{1}{2}}$ at each point $\boldsymbol{x}$ on the grid, and draw an ellipse centered at $\boldsymbol{x}$, whose axes and radii are the resulting eigenvectors and the corresponding eigenvalues, respectively. These ellipses represent the *discrimination thresholds* of the metric across space, which are inversely proportional to its local sensitivities. In other words, the ellipses illustrate the directions that require larger perturbations to induce the same change in distance. Figure 2 shows that the discrimination thresholds align with the direction of the local covariance, *i.e.*, the metric $\boldsymbol{G}(\boldsymbol{x}, \Gamma)$ behaves like a *locally adaptive* Mahalanobis metric. We also note that $\boldsymbol{G}(\boldsymbol{x}, \Gamma)$ is more sensitive in the

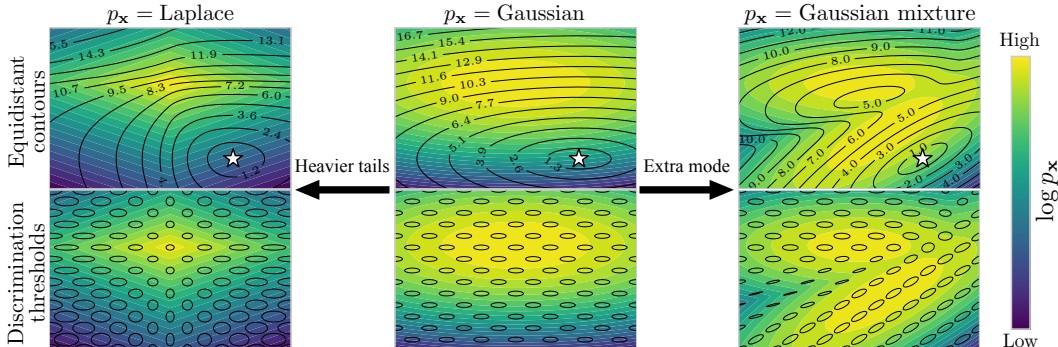

Figure 2: **Illustrating the global and local geometry of the Information-Estimation Metric (IEM) on three different prior densities.** *Top row:* Equidistant IEM contours relative to an example reference point (white star). When $p_{\mathbf{x}}$ is Gaussian (middle column), the IEM coincides with the well-known Mahalanobis distance. For a separable Laplacian prior (left column), the equidistant contours cluster and curve around the axes, following the high-probability ridges. For a Gaussian mixture prior (right column), the contours reflect the shapes of the modes. These examples illustrate how the IEM adapts to the global geometry of the given prior density. *Bottom row:* Ellipses representing the local discrimination thresholds of the local Riemannian metric $\boldsymbol{G}(\boldsymbol{x}, \Gamma)$ (Eq. (5)). Larger ellipse radii correspond to higher discrimination thresholds, *i.e.*, lower sensitivity to local perturbations. For the Gaussian prior, the local metric is constant across the entire domain (identical to the Mahalanobis metric). For the Laplace (heavy-tailed) prior, the discrimination thresholds are smaller in high-probability regions—consistent with human perception and predictions of efficient coding theories. Moreover, the orientations of the ellipses align with the equiprobable log-density contours, implying that $\boldsymbol{G}(\boldsymbol{x}, \Gamma)$ is more sensitive to perturbations that yield a larger change in the probability of $\boldsymbol{x}$. For the Gaussian mixture density, the discrimination thresholds are smaller between the modes, and the major axes of the ellipses align with the direction of larger local variance. Overall, these examples illustrate that $\boldsymbol{G}(\boldsymbol{x}, \Gamma)$ is more sensitive in regions of higher log-density curvature and to perturbations that induce larger local changes in probability.

local minima of probability in between the modes, as illustrated by the ellipses with smaller radii (smaller discrimination thresholds). This is consistent with Thm. 2, as the signals lying between modes incur large denoising errors due to the uncertainty about the mode they belong to.

**Laplace prior.** To further illustrate the influence of the density's curvature on the IEM, we now consider the case where $p_{\mathbf{x}}$ is a two-dimensional Laplace distribution, formed by taking the product of one-dimensional Laplace densities. As shown in Fig. 2, this prior density induces discrimination thresholds that increase as they move away from the high-probability ridges that lie along the axes. From the point of view of Thm. 2, this reflects the fact that the Hessian matrices $\nabla^2 \log p_{\mathbf{y}_\gamma}(\gamma \boldsymbol{x} + \mathbf{w}_\gamma)$ (specifically, their negative eigenvalues) decrease in magnitude away from the axes. For this sparse and heavy-tailed distribution, curvature is correlated with probability, so that discrimination thresholds are larger in low-probability regions, consistent with predictions from prior work on efficient coding (Ganguli & Simoncelli, 2014). From the global behavior of the distance, we also see that the equidistant contour lines tend to cluster around the axes: under a sparse prior such as the Laplace density, flipping the sign of one or several coordinates of $\boldsymbol{x}$ (landing on the other side of the high-probability ridge) incurs a large cost as measured by the IEM.

Additional illustrative examples on one-dimensional prior densities are provided in App. E.4.

## 2.3 GENERALIZED INFORMATION-ESTIMATION METRIC

The IEM is defined as the expected quadratic variation of the Itô process

$$\mathbf{z}_\gamma(\boldsymbol{x}_1, \boldsymbol{x}_2) \coloneqq \log \left( \frac{p_{\mathbf{y}_\gamma}(\gamma \boldsymbol{x}_1 + \mathbf{w}_\gamma)}{p_{\mathbf{y}_\gamma}(\gamma \boldsymbol{x}_2 + \mathbf{w}_\gamma)} \right). \qquad (9)$$

Note that the quadratic variation of $\mathbf{z}_\gamma$ is unaffected by additive *shifts* of the process (the drift coefficient is ignored). Namely, $\mathbf{z}_\gamma$ may have small or large average values while yielding the same

quadratic variation. It is therefore interesting to take such shifts into account by quantifying the deviation of $\mathbf{z}_\gamma$ from zero. A natural way to achieve this is to measure the quadratic variation of some scalar function $f(\mathbf{z}_\gamma, \gamma)$ that increases with $|\mathbf{z}_\gamma|$, thereby generalizing the IEM. When $f$ is twice differentiable, Itô's lemma shows that the diffusion coefficient of the process $f(\mathbf{z}_\gamma, \gamma)$ equals that of $\mathbf{z}_\gamma$ multiplied by $f'(\mathbf{z}_\gamma, \gamma)$—the derivative of $f(\cdot, \gamma)$ w.r.t. the first argument. We thus define:

**Definition 2.** *For any twice differentiable scalar function $f$, the generalized* IEM *is defined as*

$$\mathrm{IEM}_f(\boldsymbol{x}_1, \boldsymbol{x}_2, \Gamma) := \left( \int_0^\Gamma \mathbb{E}\left[ f'(\mathbf{z}_\gamma, \gamma)^2 \big\| \nabla \log p_{\mathbf{y}_\gamma}(\gamma \boldsymbol{x}_1 + \mathbf{w}_\gamma) - \nabla \log p_{\mathbf{y}_\gamma}(\gamma \boldsymbol{x}_2 + \mathbf{w}_\gamma) \big\|^2 \right] \mathrm{d}\gamma \right)^{\frac{1}{2}}$$

*where the expectations are taken over $p_{\mathbf{w}_\gamma}$ for each $\gamma$.*

For $f(\mathbf{z}_\gamma, \gamma) = \mathbf{z}_\gamma$, $\mathrm{IEM}_f$ recovers the IEM from Def. 1. Moreover, when $f(\mathbf{z}_\gamma, \gamma)$ satisfies $f'(\mathbf{z}_\gamma, \gamma) = 0$ if and only if $\mathbf{z}_\gamma = 0$, we have $\mathrm{IEM}_f(\boldsymbol{x}_1, \boldsymbol{x}_2) = 0$ if and only if $\boldsymbol{x}_1 = \boldsymbol{x}_2$ (positive definiteness). Unlike the IEM, however, the $\mathrm{IEM}_f$ is generally not a proper metric, as it may violate the symmetry or the triangle inequality axioms. This may or may not be considered a limitation, depending on the intended application of the distance. Definition 2 can be extended to any *non-anticipative* functional $f(\{\mathbf{z}_{\gamma'}\}_{\gamma'=0}^\gamma, \gamma)$ whose input is the entire history of the process $\mathbf{z}_{\gamma'}$ up to SNR $\gamma$. In this case, $f'$ is a Dupire derivative (Dupire, 2009; Cont & Fournie, 2010).

In App. C.4, we establish two important properties of the process $\mathbf{z}_\gamma$, which are inherited by the family of IEMs. Specifically, we show that $\mathbf{z}_\gamma$ is invariant under Euclidean isometries, *i.e.*, it is invariant to the choice of orthonormal coordinate system. Moreover, $\mathbf{z}_\gamma$ is invariant to sufficient statistics of $\mathbf{y}_\gamma$, a property that the IEMs share with the Fisher information metric (Chentsov, 1981).

## 3 EXPERIMENTS

We assess how well our proposed distances predict human judgments of similarity between photographic images. Specifically, we evaluate the IEMs on pairs of images taken from databases of psychophysical experiments, and compare the predicted distances with human similarity ratings. Computing the IEMs requires access to the score function $\nabla \log p_{\mathbf{y}_\gamma}$, or equivalently to an MMSE estimator $\mathbb{E}[\mathbf{x} \,|\, \mathbf{y}_\gamma]$, at each SNR level $\gamma$. We approximate this estimator with a learned neural denoiser $D_\theta(\mathbf{y}_\gamma, \gamma)$, which is trained to predict $\mathbf{x}$ from $(\mathbf{y}_\gamma, \gamma)$ by minimizing MSE (similarly to *unconditional* diffusion models). To evaluate our distance functions, we plug the trained denoiser into Defs. 1 and 2 and solve the integral numerically (see App. E.1 for more details).

### 3.1 IMPLEMENTATION

**Neural denoiser architecture.** We use the Hourglass Diffusion Transformer (HDiT) (Crowson et al., 2024) as our denoiser model because it can be trained efficiently and scales linearly with image resolution. We train a denoiser model from scratch on the ImageNet-1k (Deng et al., 2009) dataset, cropping the images to size $256 \times 256$. We follow most of the implementation choices of Crowson et al. (2024), but use significantly smaller models and a log-uniform schedule for the noise level. Additional training details and hyperparameters are disclosed in App. E.2.

**Choosing $f$.** The generalized $\mathrm{IEM}_f$ (Def. 2) depends on the choice of the scalar function $f$, so we examine three options: (1) *Identity function:* Setting $f(\mathbf{z}_\gamma, \gamma) = \mathbf{z}_\gamma$ corresponds to our first IEM distance (Def. 1). (2) *Quadratic function:* We take $f(\mathbf{z}_\gamma, \gamma) = \mathbf{z}_\gamma^2$ and denote by $\mathrm{IEM}_{\mathrm{sq.}}$ the resulting distance. We find this simple choice sufficient to demonstrate that $\mathrm{IEM}_f$ can adapt to different types of human data by selecting an appropriate function $f$, without supervision. (3) *Learned function:* We consider learning a parameterized function $f'_\omega$ from labeled data. The purpose of this choice is to assess whether our proposed family of distances can match human perception across several kinds of psychophysical experiments simultaneously. This is a challenging problem, since the distance must adapt to both "local" distortions near the visual sensitivity threshold (*e.g.*, small additive noise) and "global" distortions (*e.g.*, images containing similar-looking textures). Moreover, this choice provides a fairer comparison with competing methods, all of which are supervised algorithms. We implement $f'_\omega$ as a simple *causal* (non-anticipative) fully-connected network, where the output at

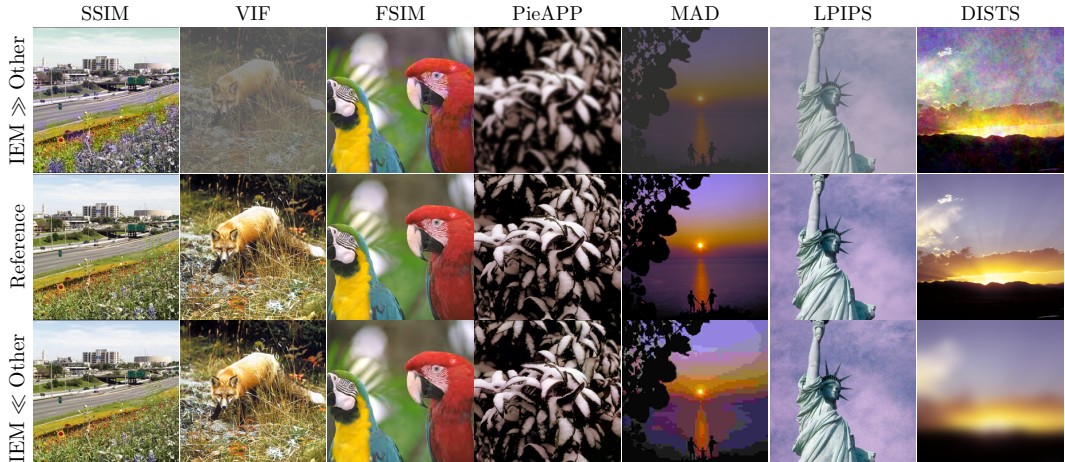

Figure 3: **Illustrating the disagreement between different types of perceptual distance measures.** We ranked the distorted images associated with each reference image in the LIVE and CSIQ databases (middle row), according to the IEM and several other metrics. Each column displays the distorted images with the largest positive (bottom row) or negative (top row) rank differences between the IEM and the compared metric (denoted in the title of the column).

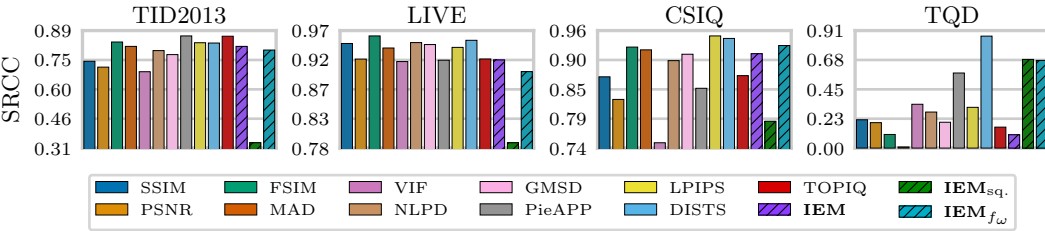

Figure 4: **Spearman's rank correlation coefficient (SRCC) results on full-reference image similarity benchmarks.** On TID2013, LIVE, and CSIQ, the IEM performs competitively with previous state-of-the-art supervised methods, but struggles on TQD (texture similarity data), as do most methods. In contrast, the unsupervised IEM$_{sq.}$ performs surprisingly well on TQD. Our supervised variant, which only learns $f_\omega$, achieves strong results on both types of databases simultaneously.

SNR $\gamma$ depends on all previous samples $\{|\mathbf{z}_{\gamma'}|\}^{\gamma}_{\gamma'=0}$. In all experiments, $f'_\omega$ is trained on data disjoint from the evaluation data. See App. E.3 for more details about learning this function.

## 3.2 PREDICTING MEAN OPINION SCORES

We evaluate our solutions using several full-reference image quality assessment databases containing mean opinion scores (MOS). In App. D.1 we report additional experiments on BAPPS (Zhang et al., 2018)—a different type of database consisting of two-alternative forced choice (2AFC) rankings.

**Common benchmarks.** We consider several standard full-reference image quality assessment benchmarks, including TID2013 (Ponomarenko et al., 2015), CSIQ (Larson & Chandler, 2010), and LIVE (Sheikh et al., 2006). Since the learned denoiser model is suited for images of size $256 \times 256$, we adjust the resolution of the images in the considered databases by first center-cropping each image to the length of its shorter edge, and then resizing it to $256 \times 256$. We compare against PSNR, SSIM (Wang et al., 2004), VIF (Sheikh & Bovik, 2006), MAD (Larson & Chandler, 2010), FSIM (Zhang et al., 2011), GMSD (Xue et al., 2014), NLPD (Laparra et al., 2016), PieAPP (Prashnani et al., 2018), LPIPS (Zhang et al., 2018), DISTS (Ding et al., 2022), and TOPIQ (Chen et al., 2024). We find that the IEM with $\Gamma = 1/4$ yields surprisingly strong results, even though it is computed solely based on denoising errors and is not exposed to human labels. Indeed, as shown in Fig. 4, this same choice of $\Gamma$ produces a strikingly high Spearman's rank correlation coefficient

(SRCC) with the human MOS across all of the aforementioned datasets. Additional performance measures demonstrate similar trends, so we report them in Figs. 7 and 8 in App. D.2.

To illustrate the differences between the IEM and the compared distance measures, we present in Fig. 3 several example images for which the IEM rankings differ the most from those of the compared methods. For each reference image in the dataset, we rank all of its distorted counterparts according to each distance measure. We then compute the difference between the ranks assigned by the IEM and those assigned by each compared method. From these differences, we take the maximum and minimum values, and sum their absolute magnitudes to quantify the degree of disagreement. Finally, we display the reference and distorted images that achieve the largest disagreement. This systematic procedure for comparing image similarity models on a given dataset is *analogous* to the maximum differentiation competition (Wang & Simoncelli, 2008). The results show that VIF, FSIM, PieAPP, and LPIPS can assign smaller distances to image pairs that are perceptually distinguishable, whereas the IEM correctly detects that the images are different. In comparison, MAD and DISTS tend to disagree with the IEM in cases involving perceptually noticeable distortions. For example, DISTS appears to favor noise over blur, whereas the IEM shows the opposite preference.

**Texture images.** We further evaluate our distance measures on the TQD textures dataset (Ding et al., 2022). Unlike the previously considered benchmarks, which contain general natural images, this dataset consists of texture images (*e.g.*, leaves or brick walls), paired both with visually similar textures and with distorted versions of the same texture (*e.g.*, Gaussian blur, JPEG compression). In this setting, human observers are expected to judge two images of the same texture as more similar to each other than a clean and distorted pair. Thus, the TQD benchmark assesses whether a perceptual distance measure produces scores consistent with human perception even when the compared images are substantially different in terms of their Euclidean distance. As in the previous experiments, we use the $256 \times 256$ denoiser model with the same preprocessing to resize the images.

As shown in Fig. 4, we find that our distance $\text{IEM}_{\text{sq.}}$ with $\Gamma = 10^6$ outperforms all other methods, except for DISTS, which was explicitly designed to handle texture images. However, $\text{IEM}_{\text{sq.}}$ does not perform well on the TID2013, LIVE, and CSIQ datasets. This highlights the flexibility of the $\text{IEM}_f$ to accommodate very different types of distortions by choosing $f$. An important question, then, is whether a single mapping $f$ can realize both types of functions. The answer is positive: our learned distance $\text{IEM}_{f_\omega}$ achieves strong performance across all datasets simultaneously, indicating the significance of $f$ and the flexibility of our distances in practice.

## 3.3 MAXIMUM DIFFERENTIATION COMPETITION AGAINST THE PSNR MEASURE

To further demonstrate the behavior of the IEM, we conduct a maximum differentiation competition against PSNR. Specifically, we minimize or maximize each perceptual distance (IEM, DISTS, etc.) via *projected* gradient descent or ascent, respectively, where the projection step during optimization constrains the PSNR of the distorted image to a fixed, pre-determined level. Figure 5 illustrates this experiment and compares the optimized images produced by the IEM and DISTS, using a PSNR constraint of 10dB. Comparisons with additional perceptual distance measures on varying levels of PSNR constraints are shown in Figs. 9 to 12. Implementation details are disclosed in App. D.3.

Interestingly, we find that minimizing the IEM consistently yields high perceptual quality images that preserve the overall geometric structure of the reference image, even under very low PSNR constraints (see Figs. 5 and 9 to 12). In contrast, minimizing other perceptual metrics produces images with unnatural artifacts, even for relatively high PSNR constraints (*e.g.*, 25dB). These results suggest that, unlike previous perceptual metrics, the IEM may serve as a stand-alone robust optimization objective (*e.g.*, for solving inverse problems), and we encourage future work to explore this potential. Furthermore, maximizing the IEM reveals that it is most sensitive to unstructured noise perturbations that push an image off the "data support" (decreasing the image's probability), consistent with our theoretical analysis in Sec. 2.

## 4 DISCUSSION

We have introduced the Information-Estimation Metric (IEM), a novel form of distance measure induced by the geometry of an underlying probability density, and provided a means of learning this

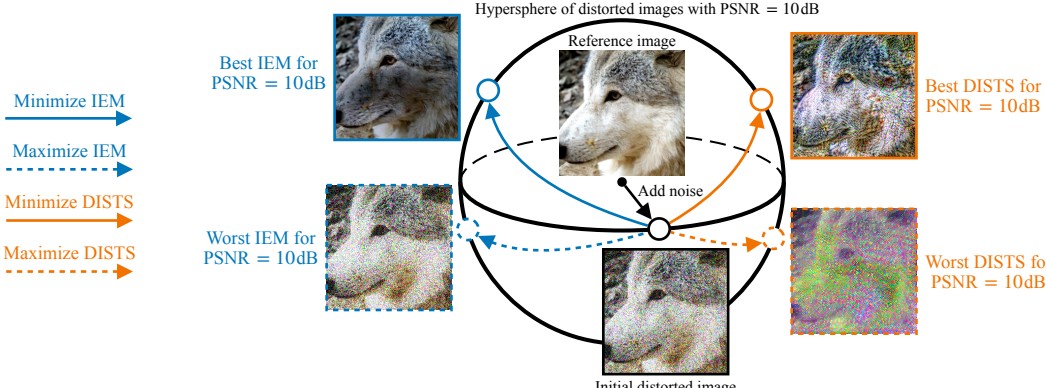

Figure 5: **Visual illustration of the maximum differentiation competition.** We corrupt a given reference image with random noise to produce a distorted image with PSNR = 10dB. This distorted image lies on the surface of a hypersphere in $\mathbb{R}^d$ centered at the reference image, with radius equal to the Euclidean norm of the added noise, as illustrated above. Starting from the distorted image, we then minimize or maximize the perceptual distance to the reference image while constraining the PSNR to 10dB. Even under such a restrictive constraint, minimizing the IEM yields artifact-free images that are perceptually similar to the reference image, whereas minimizing state-of-the-art supervised perceptual metrics such as DISTS yields unrealistic images with noticeable artifacts. Furthermore, the results obtained by maximizing the IEM support our theoretical analysis in Sec. 2, demonstrating that the IEM is most sensitive to unstructured distortions (*e.g.*, additive noise) that perturb the reference image outside the "data support."

(unsupervised) metric from samples. The definition of the IEM relies on fundamental principles that link the probability density with its local geometry, namely the pointwise I-MMSE and Tweedie–Miyasawa formulas. This relationship between probability and geometry arises from the choice of an estimation problem, in our case Gaussian denoising. Different choices of the estimation problem may yield different types of IEMs. For instance, it is possible to define an IEM using the pointwise I-MMSE relation for Poisson channels (Jiao et al., 2013), and the empirical Bayes relation for Poisson denoising (Raphan & Simoncelli, 2011). We leave these as opportunities for future work. Furthermore, the IEM does not assume that the density is supported on a low-dimensional manifold, in contrast to manifold learning approaches (see App. A.1 for further discussion). In fact, the IEM is well-defined for any valid probability density. We proved that the IEM is a valid distance metric and analyzed its local and global properties through both theoretical results and illustrative examples. To demonstrate the value of our proposed framework, we trained an IEM on the ImageNet database and found that it aligns surprisingly well with human judgments of image similarity.

The proposed IEMs (Defs. 1 and 2) require choosing a scalar hyperparameter $\Gamma$. This hyperparameter sets the maximum SNR over which the integral is computed, effectively controlling the finest resolution at which the metric is adapted to the density. Such a hyperparameter should presumably be chosen based on the fine-scale geometry of the density, or for a learned density, the complexity and size of the training set. Moreover, the generalized IEM (Def. 2) depends on the choice of the function $f$, which qualitatively controls the relative importance of log-probability ratio values compared to score differences. A systematic principle for determining both $\Gamma$ and $f$ remains an open problem. Perhaps the most important limitation of the IEM is its computational cost: Numerical estimation of the integral is more computationally demanding than evaluation of existing supervised perceptual metrics (*e.g.*, LPIPS or DISTS). This is acceptable for applications to collections of images (as in our comparison to human perceptual data), but would limit its use as an optimization objective (*e.g.*, for solving inverse problems, or optimizing compression systems). We believe that the IEM may be evaluated in a single forward pass, *e.g.*, using a strategy similar to Guth et al. (2025).

There are many potential future applications for the IEM framework. For example, it offers new opportunities for unsupervised data clustering (as illustrated in App. D.4), information retrieval, evaluating (or optimizing) image restoration and compression engines, and discriminating between generative models (using the mismatched IEM proposed in App. B). It is also natural to consider applying the principles presented in this paper to other forms of continuous signals, such as audio.

## REPRODUCIBILITY STATEMENT

Section 3 and Apps. D and E provide all the details necessary to reproduce our results, including the training hyperparameters of the denoiser model used in the computation of the IEM, the implementation details of the learned function $f_\omega$, the data preprocessing procedures, and the maximum differentiation competition experiments. Our code is available online at https://github.com/ohayonguy/information-estimation-metric.

## ACKNOWLEDGMENTS

We thank our colleagues from the Center for Computational Neuroscience at the Flatiron Institute for their helpful comments and suggestions. G.O. gratefully acknowledges the Viterbi Fellowship from the Faculty of Electrical and Computer Engineering at the Technion.

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

# Appendix

## A   RELATED WORK AND BACKGROUND

### A.1   RELATED WORK

**Metric learning.** Metric learning methods aim to learn a distance or similarity function from data that is suited to a particular clustering task (*e.g.*, face verification or recognition, image or text retrieval), such that points from the same cluster are considered closer to each other than points from different clusters (Xing et al., 2002; Kulis, 2013). Thus, in metric learning, the metric is, by design, informed either by a specific downstream task or by some other specification of the desired clustering structure within the signal domain. In contrast, the IEM we introduce in this paper is not informed by any downstream task, but is rather derived directly from the probability distribution of the (unlabeled) data. Several approaches in the literature rely on self-supervised learned representations, such as the latent space of generative models, in order to induce a Riemannian metric in the input space (Arvanitidis et al., 2017). In contrast, our approach does not rely on an explicit latent space and does not require computing geodesic distances.

Another path to metric learning is through dimensionality reduction, where the metric is derived by measuring distances (*e.g.*, Euclidean) in a low-dimensional embedding space. For example, diffusion maps (Coifman & Lafon, 2006) aim to reveal the manifold structure of data. Given a

similarity matrix and its corresponding graph, this method characterizes the geometry of the underlying manifold by simulating random walks. Computing the eigenvectors of the corresponding diffusion operator provides coordinates for embedding data into a lower-dimensional space, where Euclidean distances correspond to distances along the manifold and are referred to as "diffusion distances." However, constructing the similarity matrix requires the choice of a kernel in data space, and the diffusion "time" needs to be carefully calibrated (Shan & Daubechies, 2022). In contrast, the IEM does not make a manifold assumption and instead relies on denoising, which corresponds to reversing a diffusion process in the full signal space.

**Defining a Riemannian metric through the score of the blurred density.** Recent studies proposed different ways to define a Riemannian metric using the score of the blurred density (*i.e.*, the density of noise-corrupted signals) (Saito & Matsubara, 2025; Azeglio & Bernardo, 2025). These Riemannian metrics are then used to compute geodesics, which can, *e.g.*, be applied to interpolate between a given pair of images. Such approaches differ from our proposed IEM framework in several important ways. First, the IEM is, by definition, a global distance function from which we derive a local Riemannian metric, rather than the other way around. In fact, the IEM is not the geodesic distance corresponding to the associated Riemannian metric we derive (Eq. (5)). Second, the Riemannian metrics proposed in (Saito & Matsubara, 2025; Azeglio & Bernardo, 2025) are defined based on the score at a single blur level $\gamma$, whereas the IEM (and its associated Riemannian metric) integrates over a range of blur levels $\gamma \in [0, \Gamma]$. Third, the IEM is constructed from first principles and satisfies several important properties, *e.g.*, it reduces to the well-known Mahalanobis distance when the prior density is Gaussian.

**Information geometry.** Information geometry (Amari & Nagaoka, 2000) uses tools from differential geometry to analyze statistical models. In particular, it considers probability distributions as points lying on a Riemannian manifold whose metric is derived from the KL divergence. In the case of a family of conditional distributions $\{p_{\mathbf{y}|\mathbf{x}}(\cdot|\boldsymbol{x}) \,|\, \boldsymbol{x} \in \mathbb{R}^d\}$, this metric is given by the Fisher information matrix (Fisher, 1922), which quantifies the amount of information that $\mathbf{y}$ carries about $\mathbf{x}$ (here, $\mathbf{x}$ is considered as an unknown "parameter"). Specifically, the Fisher information matrix is defined as

$$\mathcal{I}(\boldsymbol{x}) = \mathbb{E}\big[\nabla_{\boldsymbol{x}} \log p_{\mathbf{y}|\mathbf{x}}(\mathbf{y}|\boldsymbol{x}) \nabla_{\boldsymbol{x}} \log p_{\mathbf{y}|\mathbf{x}}(\mathbf{y}|\boldsymbol{x})^\top\big], \tag{10}$$

where the expectation is taken over $p_{\mathbf{y}|\mathbf{x}}(\cdot|\boldsymbol{x})$. This metric can be used to define a *geodesic* distance in the domain of $p_{\mathbf{x}}$ (although different $\boldsymbol{x}$'s can also be compared directly with the KL divergence between the associated conditional distributions). Note that the Fisher information metric is derived solely from the given observation model, namely the *representation*, $p_{\mathbf{y}|\mathbf{x}}$, which can be completely unrelated to the prior $p_{\mathbf{x}}$ (or otherwise, one has to specify explicitly how $p_{\mathbf{y}|\mathbf{x}}$ depends on $p_{\mathbf{x}}$). Our approach departs from this classical framework in two ways (although there are qualitative analogies, see paragraph on invariance to sufficient statistics in App. C.4). First, the IEM depends directly on the prior $p_{\mathbf{x}}$, rather than through a potentially prior-dependent observation model $p_{\mathbf{y}|\mathbf{x}}$. Second, the IEM is a *global* distance metric from which we derive the local Riemannian metric $\boldsymbol{G}$ (Eqs. (5) and (6)), but this is a one-way relationship: the IEM is not a geodesic distance.

**Deriving a metric from a prior using Jeffreys rule.** Solving a Bayesian inference problem requires both a likelihood function $p_{\mathbf{y}|\mathbf{x}}$ and a prior $p_{\mathbf{x}}$. However, in some cases only the likelihood is available. In such situations, it is natural to choose a non-informative prior using Jeffreys rule (Jeffreys, 1946), which states that the prior should be proportional to the square root of the determinant of the Fisher information matrix. More relevant to our case, this relationship has also been applied in the reverse direction, where the prior is known but the likelihood is not. In particular, it has been shown that a likelihood for which the Jeffreys prior matches the data distribution satisfies the principles of efficient coding (Ganguli & Simoncelli, 2014). The reverse use of Jeffreys prior has also been explored in machine learning. For example, Lebanon (2002) considered a Riemannian metric under which the data is uniformly distributed. Since the prior density is a scalar function, they assumed an isotropic metric of the form $\boldsymbol{M}(\boldsymbol{x}) \propto \lambda(\boldsymbol{x})\boldsymbol{I}$, where $\lambda(\boldsymbol{x}) \propto p(\boldsymbol{x})^{2/d}$, and $\boldsymbol{x} \in \mathbb{R}^d$. In both of these cases, the result depends only on a scalar quantity and therefore cannot account for the varying magnitudes of discrimination thresholds in different perturbation directions (Berardino et al., 2017). In contrast, the IEM builds on the local geometry of the data distribution to define a distance that captures its anisotropic structure.

## A.2 ORIGINS OF THE POINTWISE I-MMSE FORMULA

**I-MMSE.** The I-MMSE relation (Guo et al., 2005), which is closely related to de Bruijn's identity from the 1950s (Stam, 1959), is a fundamental connection between information theory and estimation theory for Gaussian noise channels. Specifically, the I-MMSE formula relates the mutual information between $\mathbf{x}$ and $\mathbf{y}_\gamma$ to the *integrated* MMSE achievable when estimating $\mathbf{x}$ from the noisy channel. Formally, letting $I(\mathbf{x}, \mathbf{y}_\gamma)$ denote the mutual information between $\mathbf{x}$ and $\mathbf{y}_\gamma$, the I-MMSE formula (Guo et al., 2005) states that

$$I(\mathbf{x}, \mathbf{y}_\Gamma) := \mathbb{E}\left[\log\left(\frac{p_{\mathbf{y}_\Gamma|\mathbf{x}}(\mathbf{y}_\Gamma|\mathbf{x})}{p_{\mathbf{y}_\Gamma}(\mathbf{y}_\Gamma)}\right)\right] = \frac{1}{2}\int_0^\Gamma \mathbb{E}\left[\|\mathbf{x} - \mathbb{E}[\mathbf{x}|\mathbf{y}_\gamma]\|^2\right]\mathrm{d}\gamma, \tag{11}$$

where the expectation on the left-hand side is taken over the joint distribution $p_{\mathbf{x},\mathbf{y}_\Gamma}$, while on the right-hand side it is taken over $p_{\mathbf{x},\mathbf{y}_\gamma}$ for each $\gamma$. The result above holds for any Gaussian channel with SNR $\gamma$, and not only for the channel defined in Eq. (1).

**Pointwise I-MMSE.** By interchanging the order of expectation and integration on the right-hand side of Eq. (11), we obtain

$$\mathbb{E}\left[\log\left(\frac{p_{\mathbf{y}_\Gamma|\mathbf{x}}(\mathbf{y}_\Gamma|\mathbf{x})}{p_{\mathbf{y}_\Gamma}(\mathbf{y}_\Gamma)}\right)\right] = \mathbb{E}\left[\frac{1}{2}\int_0^\Gamma \|\mathbf{x} - \mathbb{E}[\mathbf{x}|\mathbf{y}_\gamma]\|^2\mathrm{d}\gamma\right]. \tag{12}$$

This reformulation highlights that the two sides of the I-MMSE formula correspond to random variables that are equal in expectation. Venkat & Weissman (2012) showed that these random variables satisfy the *pointwise* I-MMSE formula

$$\log\left(\frac{p_{\mathbf{y}_\Gamma|\mathbf{x}}(\mathbf{y}_\Gamma|\mathbf{x})}{p_{\mathbf{y}_\Gamma}(\mathbf{y}_\Gamma)}\right) = \int_0^\Gamma (\mathbf{x} - \mathbb{E}[\mathbf{x}|\mathbf{y}_\gamma]) \cdot \mathrm{d}\mathbf{w}_\gamma + \frac{1}{2}\int_0^\Gamma \|\mathbf{x} - \mathbb{E}[\mathbf{x}|\mathbf{y}_\gamma]\|^2\mathrm{d}\gamma, \tag{13}$$

where this equality holds with probability one (i.e., almost surely). As noted by Venkat & Weissman (2012), taking expectations in Eq. (2) immediately recovers the original I–MMSE formula. Indeed, the stochastic integral on the right-hand side of Eq. (2) is a martingale with zero mean, while the left-hand side corresponds to the pointwise mutual information between $\mathbf{x}$ and $\mathbf{y}_\gamma$, whose expectation yields the mutual information between these two random vectors. Using the fact that

$$p_{\mathbf{y}_\Gamma|\mathbf{x}}(\mathbf{y}_\Gamma|\mathbf{x}) = p_{\mathbf{w}_\Gamma}(\mathbf{w}_\Gamma), \tag{14}$$

it is straightforward to see that the pointwise I-MMSE equation in Eq. (13) is equivalent to Eq. (2).

# B MISMATCHED IEM AND LOCAL DECOMPOSITION OF KULLBACK–LEIBLER DIVERGENCE

Here, we show that the IEM compares the local behavior of the density in a way that resembles a "local KL divergence." This is formalized through a direct relationship between the average squared IEM between a point $\mathbf{x}$ and its additive perturbation $\tilde{\mathbf{x}} = \mathbf{x} - \boldsymbol{s}$ in a fixed direction $\boldsymbol{s}$, and the KL divergence between the distributions $p_{\mathbf{x}}$ and $p_{\tilde{\mathbf{x}}}$. To motivate this relationship, it is helpful to first introduce a generalization of the IEM.

**Information-Estimation Metric between samples from different distributions.** While the IEM from Sec. 2 is a distance function associated with a single distribution $p_{\mathbf{x}}$, it can be straightforwardly generalized to the case where $\boldsymbol{x}_1$ and $\boldsymbol{x}_2$ are assumed to come from two different distributions $p_{\mathbf{x}_1}$ and $p_{\mathbf{x}_2}$, respectively. We refer to this generalization as the *mismatched IEM*, analogously to the term "mismatched estimation" that appears in the information-estimation relations literature (Guo et al., 2013).

Our construction mirrors that of Sec. 2. We define the mismatched IEM as the expected quadratic variation of the log-probability ratio $\log\big(p_{\mathbf{y}_{1,\gamma}}(\gamma\boldsymbol{x}_1 + \mathbf{w}_\gamma)/p_{\mathbf{y}_{2,\gamma}}(\gamma\boldsymbol{x}_2 + \mathbf{w}_\gamma)\big)$, where $p_{\mathbf{y}_{i,\gamma}}$ is the distribution of $\mathbf{y}_{i,\gamma} = \gamma\mathbf{x}_i + \mathbf{w}_\gamma$. This log-probability ratio involves taking the difference between two denoisers corresponding to the two priors $p_{\mathbf{x}_1}$ and $p_{\mathbf{x}_2}$. Similarly to Sec. 2, these denoising errors can be expressed in terms of the gradients of $\log p_{\mathbf{y}_{i,\gamma}}$. Specifically,

**Definition 3.** *The* mismatched IEM *induced by the densities $p_{\mathbf{x}_1}$ and $p_{\mathbf{x}_2}$ is defined as*

$$\mathrm{IEM}_{p_{\mathbf{x}_1}, p_{\mathbf{x}_2}}(\boldsymbol{x}_1, \boldsymbol{x}_2, \Gamma) := \left( \int_0^{\Gamma} \mathbb{E}\Big[ \big\| \nabla \log p_{\mathbf{y}_{1,\gamma}}(\gamma \boldsymbol{x}_1 + \mathbf{w}_\gamma) - \nabla \log p_{\mathbf{y}_{2,\gamma}}(\gamma \boldsymbol{x}_2 + \mathbf{w}_\gamma) \big\|^2 \Big] \mathrm{d}\gamma \right)^{\frac{1}{2}}$$

*where the expectations are taken over $p_{\mathbf{w}_\gamma}$ for each $\gamma$.*

Intuitively, the mismatched IEM compares the local geometry of $\log p_{\mathbf{x}_1}$ in the vicinity of $\boldsymbol{x}_1$ with the local geometry of $\log p_{\mathbf{x}_2}$ in the vicinity of $\boldsymbol{x}_2$. When $p_{\mathbf{x}_1} = p_{\mathbf{x}_2}$, we trivially recover the IEM given in Def. 1.

**Relation to Kullback–Leibler divergence.** The mismatched IEM distance is directly related to the KL divergence between $p_{\mathbf{x}_1}$ and $p_{\mathbf{x}_2}$. Indeed, Venkat & Weissman (2012) showed that this KL divergence can be expressed as the expected quadratic variation of the log-probability ratio when the two distributions are evaluated at the *same* point, $\log\big(p_{\mathbf{y}_{1,\gamma}}(\gamma \mathbf{x}_1 + \mathbf{w}_\gamma)/p_{\mathbf{y}_{2,\gamma}}(\gamma \mathbf{x}_1 + \mathbf{w}_\gamma)\big)$, yielding

$$D_{\mathrm{KL}}(p_{\mathbf{x}_1} \,\|\, p_{\mathbf{x}_2}) = \frac{1}{2} \int_0^\infty \mathbb{E}\Big[ \big\| \nabla \log p_{\mathbf{y}_{1,\gamma}}(\gamma \mathbf{x}_1 + \mathbf{w}_\gamma) - \nabla \log p_{\mathbf{y}_{2,\gamma}}(\gamma \mathbf{x}_1 + \mathbf{w}_\gamma) \big\|^2 \Big] \mathrm{d}\gamma, \quad (15)$$

where the average is taken over $p_{\mathbf{x}_1} p_{\mathbf{w}_\gamma}$. This result also appeared in Verdú (2010). It immediately follows that the KL divergence can be expressed as the average mismatched IEM with $\Gamma = \infty$, as follows:

$$D_{\mathrm{KL}}(p_{\mathbf{x}_1} \,\|\, p_{\mathbf{x}_2}) = \frac{1}{2} \mathbb{E}\Big[ \mathrm{IEM}_{p_{\mathbf{x}_1}, p_{\mathbf{x}_2}}^2(\mathbf{x}_1, \mathbf{x}_1) \Big]. \quad (16)$$

Generalizing the above for $\Gamma < \infty$ is trivial, yielding a KL divergence between the blurred versions of $p_{\mathbf{x}_1}$ and $p_{\mathbf{x}_2}$. Equation (16) allows us to interpret the mismatched IEM between the *same* points seen as samples coming from two *different* distributions, as a *local decomposition* of the KL divergence between these two distributions. It formalizes the intuition that the (mismatched) IEM compares the local behavior of two (potentially different) densities around the two points, since the average over $p_{\mathbf{x}_1}$ yields a global comparison (given by the KL divergence). Note that $\log(p_{\mathbf{x}_1}(\mathbf{x}_1)/p_{\mathbf{x}_2}(\mathbf{x}_1))$ also qualifies as a local decomposition in the sense that its average yields the KL divergence, but unlike the mismatched IEM, it is not a valid distance (for instance, it can take negative values). The mismatched IEM can thus be thought of as a positive-definite decomposition of the log-probability ratio.

We now reinterpret Eq. (16) in the case of the IEM given by Def. 1. Consider the scenario where the two distributions are $p_{\mathbf{x}}$ and $p_{\tilde{\mathbf{x}}}$, where $\tilde{\mathbf{x}} = \mathbf{x} - \boldsymbol{s}$ for some additive (fixed) shift $\boldsymbol{s}$. We then have $p_{\tilde{\mathbf{x}}}(\tilde{\boldsymbol{x}}) = p_{\mathbf{x}}(\tilde{\boldsymbol{x}} + \boldsymbol{s})$, and thus $\mathrm{IEM}_{p_{\mathbf{x}}, p_{\tilde{\mathbf{x}}}}(\boldsymbol{x}, \tilde{\boldsymbol{x}}) = \mathrm{IEM}(\boldsymbol{x}, \tilde{\boldsymbol{x}} + \boldsymbol{s})$. In this setting, Eq. (16) becomes

$$D_{\mathrm{KL}}(p_{\mathbf{x}} \,\|\, p_{\tilde{\mathbf{x}}}) = \frac{1}{2} \mathbb{E}\big[ \mathrm{IEM}^2(\mathbf{x}, \mathbf{x} + \boldsymbol{s}) \big]. \quad (17)$$

By taking $\boldsymbol{s} = \boldsymbol{x}_2 - \boldsymbol{x}_1$, this equation allows us to interpret $\mathrm{IEM}(\boldsymbol{x}_1, \boldsymbol{x}_2)$ as the term corresponding to $\mathbf{x} = \boldsymbol{x}_1$ in the local decomposition introduced above of the KL divergence between $p_{\mathbf{x}}$ and its translation by $\boldsymbol{x}_2 - \boldsymbol{x}_1$. Again, it formalizes the intuition that the IEM compares the local behavior of the density around the two points $\boldsymbol{x}_1$ and $\boldsymbol{x}_2$.

## C    PROOFS

### C.1    PROOF OF THM. 1

**Theorem 1.** *For every $\Gamma > 0$, the* IEM *is a proper distance metric: it is symmetric, non-negative, equal to zero if and only if $\boldsymbol{x}_1 = \boldsymbol{x}_2$, and it satisfies the triangle inequality.*

*Proof.* We verify the four metric axioms.

**Symmetry.** Swapping $\boldsymbol{x}_1$ and $\boldsymbol{x}_2$ leaves the squared norm unchanged, so

$$\mathrm{IEM}(\boldsymbol{x}_1, \boldsymbol{x}_2, \Gamma) = \mathrm{IEM}(\boldsymbol{x}_2, \boldsymbol{x}_1, \Gamma). \quad (18)$$

**Non-negativity.** By definition, the integrand is a squared norm and is thus nonnegative. The integral and square root preserve nonnegativity, hence $\text{IEM}(\boldsymbol{x}_1, \boldsymbol{x}_2, \Gamma) \geq 0$.

**Positive definiteness.** Suppose $\text{IEM}(\boldsymbol{x}_1, \boldsymbol{x}_2, \Gamma) = 0$. Then for Lebesgue-a.e. $\gamma \in [0, \Gamma]$ we have that

$$\mathbb{E}\left[\left\|\nabla \log p_{\mathbf{y}_\gamma}(\gamma \boldsymbol{x}_1 + \mathbf{w}_\gamma) - \nabla \log p_{\mathbf{y}_\gamma}(\gamma \boldsymbol{x}_2 + \mathbf{w}_\gamma)\right\|^2\right] = 0, \tag{19}$$

which implies

$$\nabla \log p_{\mathbf{y}_\gamma}(\gamma \boldsymbol{x}_1 + \mathbf{w}_\gamma) = \nabla \log p_{\mathbf{y}_\gamma}(\gamma \boldsymbol{x}_2 + \mathbf{w}_\gamma) \quad \text{a.s. in } \mathbf{w}_\gamma. \tag{20}$$

Because $\mathbf{w}_\gamma$ has a strictly positive density on $\mathbb{R}^d$, it follows that

$$\nabla \log p_{\mathbf{y}_\gamma}(\boldsymbol{y}) = \nabla \log p_{\mathbf{y}_\gamma}(\boldsymbol{y} + \gamma(\boldsymbol{x}_1 - \boldsymbol{x}_2)) \quad \text{for Lebesgue-a.e. } \boldsymbol{y}. \tag{21}$$

Thus the function

$$g_\gamma(\boldsymbol{y}) := \log p_{\mathbf{y}_\gamma}(\boldsymbol{y} + \gamma(\boldsymbol{x}_1 - \boldsymbol{x}_2)) - \log p_{\mathbf{y}_\gamma}(\boldsymbol{y}) \tag{22}$$

is constant a.e., say $g_\gamma(\boldsymbol{y}) = c_\gamma$. Exponentiating, we obtain

$$p_{\mathbf{y}_\gamma}(\boldsymbol{y} + \gamma(\boldsymbol{x}_1 - \boldsymbol{x}_2)) = e^{c_\gamma} \, p_{\mathbf{y}_\gamma}(\boldsymbol{y}). \tag{23}$$

Integrating both sides over $\mathbb{R}^d$ gives

$$1 = \int p_{\mathbf{y}_\gamma}(\boldsymbol{y} + \gamma(\boldsymbol{x}_1 - \boldsymbol{x}_2))\mathrm{d}\boldsymbol{y} = e^{c_\gamma} \int p_{\mathbf{y}_\gamma}(\boldsymbol{y})\mathrm{d}\boldsymbol{y} = e^{c_\gamma}, \tag{24}$$

so $c_\gamma = 0$. Hence

$$p_{\mathbf{y}_\gamma}(\boldsymbol{y} + \gamma(\boldsymbol{x}_1 - \boldsymbol{x}_2)) = p_{\mathbf{y}_\gamma}(\boldsymbol{y}) \quad \text{for Lebesgue-a.e. } \boldsymbol{y}. \tag{25}$$

Fix some $\gamma > 0$ so that the above holds. This means that $p_{\mathbf{y}_\gamma}$ is invariant under translations by the vector $\gamma(\boldsymbol{x}_1 - \boldsymbol{x}_2)$. However, there is no probability density on $\mathbb{R}^d$ that is invariant under a nonzero translation. To show that this is true, if $\boldsymbol{x}_1 \neq \boldsymbol{x}_2$, consider the sets

$$B_k = \left\{\boldsymbol{y} \in \mathbb{R}^d \,\middle|\, k \leq \left\langle \boldsymbol{y}, \frac{\boldsymbol{x}_1 - \boldsymbol{x}_2}{\gamma \|\boldsymbol{x}_1 - \boldsymbol{x}_2\|^2} \right\rangle < k + 1 \right\} \tag{26}$$

for $k \in \mathbb{Z}$. $B_k$ forms a partition of $\mathbb{R}^d$, so $\sum_{k \in \mathbb{Z}} \int_{B_k} p_{\mathbf{y}_\gamma}(\boldsymbol{y})\mathrm{d}\boldsymbol{y} = 1$. But by translation invariance, the terms in the sum do not depend on $k$, which is a contradiction. Therefore the translation vector must be zero, i.e., $\gamma(\boldsymbol{x}_1 - \boldsymbol{x}_2) = 0$, which implies $\boldsymbol{x}_1 = \boldsymbol{x}_2$.

**Triangle inequality.** For each $\gamma$, define

$$\mathcal{M}_\gamma(\boldsymbol{x}_i, \boldsymbol{x}_j) := \left(\mathbb{E}\left[\left\|\nabla \log p_{\mathbf{y}_\gamma}(\gamma \boldsymbol{x}_i + \mathbf{w}_\gamma) - \nabla \log p_{\mathbf{y}_\gamma}(\gamma \boldsymbol{x}_j + \mathbf{w}_\gamma)\right\|^2\right]\right)^{1/2}. \tag{27}$$

$\mathcal{M}_\gamma$ trivially satisfies the triangle inequality:

$$\mathcal{M}_\gamma(\boldsymbol{x}_1, \boldsymbol{x}_3) \leq \mathcal{M}_\gamma(\boldsymbol{x}_1, \boldsymbol{x}_2) + \mathcal{M}_\gamma(\boldsymbol{x}_2, \boldsymbol{x}_3). \tag{28}$$

Integrating over $\gamma$ and applying the Minkowski inequality, we get

$$\begin{aligned}
\text{IEM}^2(\boldsymbol{x}_1, \boldsymbol{x}_3, \Gamma) &= \int_0^\Gamma \mathcal{M}_\gamma(\boldsymbol{x}_1, \boldsymbol{x}_3)^2 \mathrm{d}\gamma \\
&\leq \int_0^\Gamma \left(\mathcal{M}_\gamma(\boldsymbol{x}_1, \boldsymbol{x}_2) + \mathcal{M}_\gamma(\boldsymbol{x}_2, \boldsymbol{x}_3)\right)^2 \mathrm{d}\gamma \\
&\leq \left(\sqrt{\int_0^\Gamma \mathcal{M}_\gamma(\boldsymbol{x}_1, \boldsymbol{x}_2)^2 \mathrm{d}\gamma} + \sqrt{\int_0^\Gamma \mathcal{M}_\gamma(\boldsymbol{x}_2, \boldsymbol{x}_3)^2 \mathrm{d}\gamma}\right)^2.
\end{aligned} \tag{29}$$

Taking the square root on both sides gives

$$\text{IEM}(\boldsymbol{x}_1, \boldsymbol{x}_3, \Gamma) \leq \text{IEM}(\boldsymbol{x}_1, \boldsymbol{x}_2, \Gamma) + \text{IEM}(\boldsymbol{x}_2, \boldsymbol{x}_3, \Gamma). \tag{30}$$

$\square$

## C.2 Proof of Thm. 2

**Theorem 2.** *The local Riemannian metric derived from the second-order Taylor expansion of the squared* IEM *is given by*

$$\boldsymbol{G}(\boldsymbol{x}, \Gamma) = \int_0^\Gamma \gamma^2 \mathbb{E}\Big[\big(\nabla^2 \log p_{\mathbf{y}_\gamma}(\gamma \boldsymbol{x} + \mathbf{w}_\gamma)\big)^2\Big] \mathrm{d}\gamma \tag{5}$$

$$= \int_0^\Gamma \mathbb{E}\Big[\big(\boldsymbol{I} - \gamma \mathrm{Cov}[\mathbf{x} \,|\, \mathbf{y}_\gamma = \gamma \boldsymbol{x} + \mathbf{w}_\gamma]\big)^2\Big] \mathrm{d}\gamma, \tag{6}$$

*where the expectations are taken over* $p_{\mathbf{w}_\gamma}$ *for each* $\gamma$. *Moreover, for* $\Gamma = \infty$ *we have*

$$\mathbb{E}[\boldsymbol{G}(\mathbf{x})] = \mathbb{E}\big[-\nabla^2 \log p_{\mathbf{x}}(\mathbf{x})\big] = \mathbb{E}\big[\nabla \log p_{\mathbf{x}}(\mathbf{x}) \nabla \log p_{\mathbf{x}}(\mathbf{x})^\top\big], \tag{7}$$

*where we denote* $\boldsymbol{G}(\boldsymbol{x}) := \boldsymbol{G}(\boldsymbol{x}, \infty)$ *and the expectations are taken over* $p_{\mathbf{x}}$.

*Proof.* We begin by taking the Taylor expansion of the IEM to derive a first expression for the local metric $\boldsymbol{G}(\boldsymbol{x}, \Gamma)$ in terms of the Hessian matrix of $\log p_{\mathbf{y}_\gamma}$ (App. C.2.1). We then derive an equivalent expression in terms of the covariance of the posterior $p_{\mathbf{x}|\mathbf{y}_\gamma}$ (App. C.2.2). Finally, we relate the average of the metric $\boldsymbol{G}(\mathbf{x})$ to the average of the Hessian of $\log p_{\mathbf{x}}$ (App. C.2.3).

### C.2.1 Taylor expansion of the IEM

We Taylor-expand the IEM distance between $\boldsymbol{x}$ and $\boldsymbol{x} + \boldsymbol{\epsilon}$ in $\boldsymbol{\epsilon}$:

$$\mathrm{IEM}^2(\boldsymbol{x}, \boldsymbol{x} + \boldsymbol{\epsilon}, \Gamma) = \int_0^\Gamma \mathbb{E}\Big[\big\|\nabla \log p_{\mathbf{y}_\gamma}(\gamma \boldsymbol{x} + \gamma \boldsymbol{\epsilon} + \mathbf{w}_\gamma) - \nabla \log p_{\mathbf{y}_\gamma}(\gamma \boldsymbol{x} + \mathbf{w}_\gamma)\big\|^2\Big] \mathrm{d}\gamma \tag{31}$$

$$= \int_0^\Gamma \mathbb{E}\Big[\big\|\gamma \nabla^2 \log p_{\mathbf{y}_\gamma}(\gamma \boldsymbol{x} + \mathbf{w}_\gamma)\boldsymbol{\epsilon} + o(\boldsymbol{\epsilon})\big\|^2\Big] \mathrm{d}\gamma \tag{32}$$

$$= \boldsymbol{\epsilon}^\top \left(\int_0^\Gamma \gamma^2 \mathbb{E}\Big[\big(\nabla^2 \log p_{\mathbf{y}_\gamma}(\gamma \boldsymbol{x} + \mathbf{w}_\gamma)\big)^2\Big] \mathrm{d}\gamma\right) \boldsymbol{\epsilon} + o(\|\boldsymbol{\epsilon}\|^2) \tag{33}$$

We thus have

$$\boldsymbol{G}(\boldsymbol{x}, \Gamma) = \int_0^\Gamma \gamma^2 \mathbb{E}\Big[\big(\nabla^2 \log p_{\mathbf{y}_\gamma}(\gamma \boldsymbol{x} + \mathbf{w}_\gamma)\big)^2\Big] \mathrm{d}\gamma. \tag{34}$$

### C.2.2 From the Hessian of the noisy channel log-density to the posterior covariance

We now show that the Hessian of $\log p_{\mathbf{y}_\gamma}$ can be expressed in terms of the posterior covariance of $\mathbf{x}$ conditioned on $\mathbf{y}_\gamma$:

$$\nabla^2 \log p_{\mathbf{y}_\gamma}(\boldsymbol{y}) = \mathrm{Cov}[\mathbf{x} \,|\, \mathbf{y}_\gamma = \boldsymbol{y}] - \frac{1}{\gamma}\boldsymbol{I}. \tag{35}$$

This relationship has already appeared in the literature in several contexts, and is often referred to as the "second-order Tweedie identity." To the best of our knowledge, it was first derived by Hatsell & Nolte (1971, Proposition 3). For completeness and notational consistency, we include a derivation here.

We have

$$\log p_{\mathbf{y}_\gamma}(\boldsymbol{y}) = \log\left(\int p_{\mathbf{x}}(\boldsymbol{x}) p_{\mathbf{y}_\gamma|\mathbf{x}}(\boldsymbol{y}|\boldsymbol{x}) \mathrm{d}\boldsymbol{x}\right). \tag{36}$$

Differentiating w.r.t. $\boldsymbol{y}$ gives

$$\nabla \log p_{\mathbf{y}_\gamma}(\boldsymbol{y}) = \frac{1}{p_{\mathbf{y}_\gamma}(\boldsymbol{y})} \int p_{\mathbf{x}}(\boldsymbol{x}) \nabla_{\boldsymbol{y}} p_{\mathbf{y}_\gamma|\mathbf{x}}(\boldsymbol{y}|\boldsymbol{x}) \mathrm{d}\boldsymbol{x}. \tag{37}$$

Differentiating again w.r.t. $\boldsymbol{y}$ gives

$$\nabla^2 \log p_{\mathbf{y}_\gamma}(\boldsymbol{y}) = \frac{1}{p_{\mathbf{y}_\gamma}(\boldsymbol{y})} \int p_{\mathbf{x}}(\boldsymbol{x}) \nabla_{\boldsymbol{y}}^2 p_{\mathbf{y}_\gamma|\mathbf{x}}(\boldsymbol{y}|\boldsymbol{x}) \mathrm{d}\boldsymbol{x}$$

$$- \frac{1}{p_{\mathbf{y}_\gamma}(\boldsymbol{y})^2} \left( \int p_{\mathbf{x}}(\boldsymbol{x}) \nabla_{\boldsymbol{y}} p_{\mathbf{y}_\gamma|\mathbf{x}}(\boldsymbol{y}|\boldsymbol{x}) \mathrm{d}\boldsymbol{x} \right) \left( \int p_{\mathbf{x}}(\boldsymbol{x}) \nabla_{\boldsymbol{y}} p_{\mathbf{y}_\gamma|\mathbf{x}}(\boldsymbol{y}|\boldsymbol{x}) \mathrm{d}\boldsymbol{x} \right)^\top \quad (38)$$

$$= \mathbb{E}\left[ \left. \frac{\nabla_{\boldsymbol{y}}^2 p_{\mathbf{y}_\gamma|\mathbf{x}}(\boldsymbol{y}|\mathbf{x})}{p_{\mathbf{y}_\gamma|\mathbf{x}}(\boldsymbol{y}|\mathbf{x})} \right| \mathbf{y}_\gamma = \boldsymbol{y} \right]$$

$$- \mathbb{E}\big[ \nabla_{\boldsymbol{y}} \log p_{\mathbf{y}_\gamma|\mathbf{x}}(\boldsymbol{y}|\mathbf{x}) \,\big|\, \mathbf{y}_\gamma = \boldsymbol{y} \big] \mathbb{E}\big[ \nabla_{\boldsymbol{y}} \log p_{\mathbf{y}_\gamma|\mathbf{x}}(\boldsymbol{y}|\mathbf{x}) \,\big|\, \mathbf{y}_\gamma = \boldsymbol{y} \big]^\top \quad (39)$$

Here, $\mathbf{y}_\gamma \,|\, \mathbf{x} \sim \mathcal{N}(\gamma\mathbf{x}, \gamma\boldsymbol{I})$. A direct calculation gives

$$\nabla_{\boldsymbol{y}} \log p_{\mathbf{y}_\gamma|\mathbf{x}}(\boldsymbol{y}|\boldsymbol{x}) = \boldsymbol{x} - \frac{1}{\gamma}\boldsymbol{y}, \quad \text{and} \quad (40)$$

$$\frac{\nabla_{\boldsymbol{y}}^2 p_{\mathbf{y}_\gamma|\mathbf{x}}(\boldsymbol{y}|\mathbf{x})}{p_{\mathbf{y}_\gamma|\mathbf{x}}(\boldsymbol{y}|\mathbf{x})} = \left( \boldsymbol{x} - \frac{1}{\gamma}\boldsymbol{y} \right)\left( \boldsymbol{x} - \frac{1}{\gamma}\boldsymbol{y} \right)^\top - \frac{1}{\gamma}\boldsymbol{I}. \quad (41)$$

Substituting into Eq. (39) and rearranging then yields

$$\nabla^2 \log p_{\mathbf{y}_\gamma}(\boldsymbol{y}) = \mathrm{Cov}[\mathbf{x} \,|\, \mathbf{y}_\gamma = \boldsymbol{y}] - \frac{1}{\gamma}\boldsymbol{I}. \quad (42)$$

Finally, injecting Eq. (42) into Eq. (34) gives the second expression for the local metric:

$$\boldsymbol{G}(\boldsymbol{x}, \Gamma) = \int_0^\Gamma \mathbb{E}\Big[ (\boldsymbol{I} - \gamma \mathrm{Cov}[\mathbf{x} \,|\, \mathbf{y}_\gamma = \gamma\boldsymbol{x} + \mathbf{w}_\gamma])^2 \Big] \mathrm{d}\gamma. \quad (43)$$

### C.2.3 AVERAGE LOCAL METRIC

We begin by decomposing the Hessian of the log-density $\log p_{\mathbf{x}}$. Specifically, we use Eq. (4) to express the log-probability ratio between a point $\boldsymbol{x}$ and a perturbed version of it $\boldsymbol{x} + \boldsymbol{\epsilon}$. Taking $\Gamma \to \infty$ and averaging over $p_{\mathbf{w}_\gamma}$ gives

$$\log \left( \frac{p_{\mathbf{x}}(\boldsymbol{x})}{p_{\mathbf{x}}(\boldsymbol{x} + \boldsymbol{\epsilon})} \right) = \frac{1}{2} \int_0^\infty \mathbb{E}\Big[ \|\boldsymbol{e}_\gamma(\boldsymbol{x} + \boldsymbol{\epsilon}, \mathbf{w}_\gamma)\|^2 - \|\boldsymbol{e}_\gamma(\boldsymbol{x}, \mathbf{w}_\gamma)\|^2 \Big] \mathrm{d}\gamma \quad (44)$$

Next, we take the Taylor expansion of the tracking error. From Tweedie–Miyasawa,

$$\boldsymbol{e}_\gamma(\boldsymbol{x}, \mathbf{w}_\gamma) \coloneqq \boldsymbol{x} - \mathbb{E}[\mathbf{x}|\mathbf{y}_\gamma = \gamma\boldsymbol{x} + \mathbf{w}_\gamma] = -\frac{1}{\gamma}\mathbf{w}_\gamma - \nabla \log p_{\mathbf{y}_\gamma}(\gamma\boldsymbol{x} + \mathbf{w}_\gamma), \quad (45)$$

we have

$$-\boldsymbol{e}_\gamma(\boldsymbol{x} + \boldsymbol{\epsilon}, \mathbf{w}_\gamma) = \frac{1}{\gamma}\mathbf{w}_\gamma + \nabla \log p_{\mathbf{y}_\gamma}(\gamma\boldsymbol{x} + \mathbf{w}_\gamma) + \gamma \nabla^2 \log p_{\mathbf{y}_\gamma}(\gamma\boldsymbol{x} + \mathbf{w}_\gamma)\boldsymbol{\epsilon}$$

$$+ \frac{\gamma^2}{2} \nabla^3 \log p_{\mathbf{y}_\gamma}(\gamma\boldsymbol{x} + \mathbf{w}_\gamma)(\boldsymbol{\epsilon}, \boldsymbol{\epsilon}) + o(\|\boldsymbol{\epsilon}\|^2), \quad (46)$$

where we write $\mathbf{A}(\boldsymbol{x}, \boldsymbol{y}) = (\sum_{jk} A_{ijk} x_j y_k)_i$ for the partial contraction of a symmetric third-order tensor $\mathbf{A}$ against the vectors $\boldsymbol{x}, \boldsymbol{y}$. By inserting Eq. (46) into the expression of the log-probability ratio in Eq. (44) and expanding the square, we obtain

$$\log \left( \frac{p_{\mathbf{x}}(\boldsymbol{x})}{p_{\mathbf{x}}(\boldsymbol{x} + \boldsymbol{\epsilon})} \right)$$

$$= \frac{1}{2} \int_0^\infty \mathbb{E}\Big[ 2\Big\langle \frac{1}{\gamma}\mathbf{w}_\gamma + \nabla \log p_{\mathbf{y}_\gamma}(\gamma\boldsymbol{x} + \mathbf{w}_\gamma), \gamma\nabla^2 \log p_{\mathbf{y}_\gamma}(\gamma\boldsymbol{x} + \mathbf{w}_\gamma)\boldsymbol{\epsilon} + \frac{\gamma^2}{2}\nabla^3 \log p_{\mathbf{y}_\gamma}(\gamma\boldsymbol{x} + \mathbf{w}_\gamma)(\boldsymbol{\epsilon}, \boldsymbol{\epsilon}) \Big\rangle$$

$$+ \big\| \gamma\nabla^2 \log p_{\mathbf{y}_\gamma}(\gamma\boldsymbol{x} + \mathbf{w}_\gamma)\boldsymbol{\epsilon} \big\|^2 \Big] \mathrm{d}\gamma + o(\|\boldsymbol{\epsilon}\|^2) \quad (47)$$

$$= \boldsymbol{\epsilon}^\top \int_0^\infty \mathbb{E}\Big[ \nabla^2 \log p_{\mathbf{y}_\gamma}(\gamma\boldsymbol{x} + \mathbf{w}_\gamma)\Big( \frac{1}{\gamma}\mathbf{w}_\gamma + \nabla \log p_{\mathbf{y}_\gamma}(\gamma\boldsymbol{x} + \mathbf{w}_\gamma) \Big) \Big] \gamma \mathrm{d}\gamma$$

$$+ \frac{1}{2}\boldsymbol{\epsilon}^\top \left( \int_0^\infty \mathbb{E}\Big[ (\nabla^2 \log p_{\mathbf{y}_\gamma}(\gamma\boldsymbol{x} + \mathbf{w}_\gamma))^2 + \nabla^3 \log p_{\mathbf{y}_\gamma}(\gamma\boldsymbol{x} + \mathbf{w}_\gamma)\Big( \frac{1}{\gamma}\mathbf{w}_\gamma + \nabla \log p_{\mathbf{y}_\gamma}(\gamma\boldsymbol{x} + \mathbf{w}_\gamma) \Big) \Big] \gamma^2 \mathrm{d}\gamma \right)\boldsymbol{\epsilon}$$

$$(48)$$

By identification, it follows that

$$-\nabla^2 \log p_{\mathbf{x}}(\boldsymbol{x}) = \int_0^\infty \mathbb{E}\left[\left(\nabla^2 \log p_{\mathbf{y}_\gamma}(\gamma \boldsymbol{x} + \mathbf{w}_\gamma)\right)^2 + \nabla^3 \log p_{\mathbf{y}_\gamma}(\gamma \boldsymbol{x} + \mathbf{w}_\gamma)\left(\frac{1}{\gamma}\mathbf{w}_\gamma + \nabla \log p_{\mathbf{y}_\gamma}(\gamma \boldsymbol{x} + \mathbf{w}_\gamma)\right)\right]\gamma^2 \mathrm{d}\gamma.$$
(49)

Taking the expectation over $p_{\mathbf{x}}$, we obtain

$$\mathbb{E}\left[-\nabla^2 \log p_{\mathbf{x}}(\mathbf{x})\right] = \int_0^\infty \mathbb{E}\left[\left(\nabla^2 \log p_{\mathbf{y}_\gamma}(\mathbf{y}_\gamma)\right)^2 + \nabla^3 \log p_{\mathbf{y}_\gamma}(\mathbf{y}_\gamma)\left(\frac{1}{\gamma}\mathbf{w}_\gamma + \nabla \log p_{\mathbf{y}_\gamma}(\mathbf{y}_\gamma)\right)\right]\gamma^2 \mathrm{d}\gamma,$$
(50)

where we used the fact that $\mathbf{y}_\gamma = \gamma\mathbf{x} + \mathbf{w}_\gamma$. The second term in the expectation then vanishes due to Stein's lemma (Stein, 1981),

$$\mathbb{E}\left[\nabla^3 \log p_{\mathbf{y}_\gamma}(\mathbf{y}_\gamma)\left(\frac{1}{\gamma}\mathbf{w}_\gamma\right)\right] = \mathbb{E}\left[\nabla^2 \Delta \log p_{\mathbf{y}_\gamma}(\mathbf{y}_\gamma)\right],$$
(51)

while applying integration by parts yields

$$\mathbb{E}\left[\nabla^3 \log p_{\mathbf{y}_\gamma}(\mathbf{y}_\gamma)\left(\nabla \log p_{\mathbf{y}_\gamma}(\mathbf{y}_\gamma)\right)\right] = -\mathbb{E}\left[\nabla^2 \Delta \log p_{\mathbf{y}_\gamma}(\mathbf{y}_\gamma)\right].$$
(52)

Finally, we obtain

$$\mathbb{E}\left[-\nabla^2 \log p_{\mathbf{x}}(\boldsymbol{x})\right] = \int_0^\infty \mathbb{E}\left[\left(\nabla^2 \log p_{\mathbf{y}_\gamma}(\mathbf{y}_\gamma)\right)^2\right]\gamma^2 \mathrm{d}\gamma = \mathbb{E}[\boldsymbol{G}(\mathbf{x})],$$
(53)

which is the desired relationship.

The last equality in the theorem is generic and classical. By expanding the derivatives, we have

$$-\int p_{\mathbf{x}}(\boldsymbol{x})\nabla^2 \log p_{\mathbf{x}}(\boldsymbol{x})\mathrm{d}\boldsymbol{x} = \int \left(p_{\mathbf{x}}(\boldsymbol{x})\nabla \log p_{\mathbf{x}}(\boldsymbol{x})\nabla \log p_{\mathbf{x}}(\boldsymbol{x})^\top - \nabla^2 p_{\mathbf{x}}(\boldsymbol{x})\right)\mathrm{d}\boldsymbol{x}.$$
(54)

The second term in the integral on the right-hand side vanishes through integration by parts.

$\square$

### C.3 Proof that the IEM coincides with the Mahalanobis distance for Gaussian priors

Here, we prove the following proposition:

**Proposition 1.** *Let $\mathbf{x}$ be a Gaussian random vector with mean $\boldsymbol{\mu}$ and covariance $\boldsymbol{\Sigma} \succeq 0$. For $\gamma \geq 0$, define*

$$\mathbf{y}_\gamma = \gamma\mathbf{x} + \mathbf{w}_\gamma, \quad \mathbf{w}_\gamma \sim \mathcal{N}(\mathbf{0}, \gamma\boldsymbol{I}), \quad \mathbf{w}_\gamma \perp \mathbf{x}.$$

*Then the MMSE estimator of $\mathbf{x}$ given $\mathbf{y}_\gamma$ is given by*

$$\mathbb{E}[\mathbf{x} \mid \mathbf{y}_\gamma] = \boldsymbol{\mu} + \boldsymbol{K}_\gamma(\mathbf{y}_\gamma - \gamma\boldsymbol{\mu}),$$

*where $\boldsymbol{K}_\gamma := \gamma\boldsymbol{\Sigma}(\gamma^2\boldsymbol{\Sigma} + \gamma\boldsymbol{I})^{-1}$. Further, for any $\boldsymbol{x}_1, \boldsymbol{x}_2$ with $\Delta := \boldsymbol{x}_1 - \boldsymbol{x}_2$, if $\boldsymbol{\Sigma} \succ 0$,*

$$\text{IEM}^2(\boldsymbol{x}_1, \boldsymbol{x}_2, \infty) = \int_0^\infty \|\mathbf{e}_\gamma(\boldsymbol{x}_1, \mathbf{w}_\gamma) - \mathbf{e}_\gamma(\boldsymbol{x}_2, \mathbf{w}_\gamma)\|^2 \mathrm{d}\gamma = \Delta^\top \boldsymbol{\Sigma}^{-1}\Delta,$$

*where $\mathbf{e}_\gamma(\boldsymbol{x}, \mathbf{w}_\gamma) := \boldsymbol{x} - \mathbb{E}[\mathbf{x}|\mathbf{y}_\gamma = \gamma\boldsymbol{x} + \mathbf{w}_\gamma]$. If $\boldsymbol{\Sigma} \succeq 0$ is singular, the integral equals $\Delta^\top \boldsymbol{\Sigma}^\dagger \Delta$ provided $\Delta \in \text{range}(\boldsymbol{\Sigma})$ (where $\boldsymbol{\Sigma}^\dagger$ is the pseudoinverse of $\Sigma$) and diverges to $+\infty$ otherwise.*

*Proof.* If $\mathbf{x}$ is Gaussian, then $(\mathbf{x}, \mathbf{y}_\gamma)$ is jointly Gaussian, with mean $(\boldsymbol{\mu}, \gamma\boldsymbol{\mu})$ and joint covariance

$$\begin{pmatrix} \boldsymbol{\Sigma} & \gamma\boldsymbol{\Sigma} \\ \gamma\boldsymbol{\Sigma} & \gamma^2\boldsymbol{\Sigma} + \gamma\boldsymbol{I} \end{pmatrix}$$
(55)

By elementary properties of Gaussian distributions, it follows that $\mathbf{x}$ conditioned on $\mathbf{y}_\gamma$ also has a Gaussian distribution, with mean given by

$$\mathbb{E}[\mathbf{x} \mid \mathbf{y}_\gamma] = \boldsymbol{\mu} + \boldsymbol{K}_\gamma(\mathbf{y}_\gamma - \gamma\boldsymbol{\mu}),$$
(56)

where $K_\gamma := \gamma\Sigma(\gamma^2\Sigma + \gamma I)^{-1}$. The denoising error vector is then, for any $x$,

$$\mathbf{e}_\gamma(x, \mathbf{w}_\gamma) = x - \mu - K_\gamma(\gamma(x - \mu) + \mathbf{w}_\gamma). \tag{57}$$

Hence,

$$\mathbf{e}_\gamma(x_1, \mathbf{w}_\gamma) - \mathbf{e}_\gamma(x_2, \mathbf{w}_\gamma) = (I - \gamma K_\gamma)\Delta. \tag{58}$$

Since $K_\gamma = \gamma\Sigma(\gamma^2\Sigma + \gamma I)^{-1} = \Sigma(\gamma\Sigma + I)^{-1}$, we have

$$I - \gamma K_\gamma = I - \gamma\Sigma(\gamma\Sigma + I)^{-1} = (\gamma\Sigma + I)(\gamma\Sigma + I)^{-1} - \gamma\Sigma(\gamma\Sigma + I)^{-1} = (\gamma\Sigma + I)^{-1}. \tag{59}$$

Therefore

$$\|\mathbf{e}_\gamma(x_1, \mathbf{w}_\gamma) - \mathbf{e}_\gamma(x_2, \mathbf{w}_\gamma)\|^2 = \Delta^\top(\gamma\Sigma + I)^{-2}\Delta. \tag{60}$$

Assume first $\Sigma \succ 0$ and diagonalize $\Sigma = U\Lambda U^\top$ with $\Lambda = \mathrm{diag}(\lambda_1, \ldots, \lambda_d)$, $\lambda_i > 0$. Writing $a := U^\top\Delta$,

$$\Delta^\top(\gamma\Sigma + I)^{-2}\Delta = \sum_{i=1}^d \frac{a_i^2}{(1 + \gamma\lambda_i)^2}. \tag{61}$$

Integrating termwise,

$$\int_0^\infty \frac{\mathrm{d}\gamma}{(1 + \gamma\lambda_i)^2} = \frac{1}{\lambda_i}\left[-\frac{1}{1 + \gamma\lambda_i}\right]_0^\infty = \frac{1}{\lambda_i}. \tag{62}$$

Thus,

$$\int_0^\infty \Delta^\top(\gamma\Sigma + I)^{-2}\Delta\,\mathrm{d}\gamma = \sum_{i=1}^d \frac{a_i^2}{\lambda_i} = \Delta^\top U\Lambda^{-1}U^\top\Delta = \Delta^\top\Sigma^{-1}\Delta. \tag{63}$$

If $\Sigma \succeq 0$ is singular, decompose with $\lambda_i \geq 0$ and note that the integral of $a_i^2/(1 + \gamma\lambda_i)^2$ diverges when $\lambda_i = 0$ and $a_i \neq 0$, while it equals 0 when $a_i = 0$. Thus finiteness requires $\Delta \in \mathrm{range}(\Sigma)$, in which case the same computation over the nonzero eigenvalues yields $\Delta^\top\Sigma^\dagger\Delta$. $\square$

## C.4 ADDITIONAL PROPERTIES OF THE PROCESS $\mathbf{z}_\gamma$

**Invariance to Euclidean isometries.** In Sec. 2.3 we show that one may derive different kinds of distance functions from the process $\mathbf{z}_\gamma$. A natural question, then, is whether such distances are preserved under Euclidean isometries. In other words, do these distance functions depend on the coordinate system in which $\mathbf{x}$ is represented? The following proposition establishes that the process $\mathbf{z}_\gamma$ is invariant to such isometries, in the sense that changing the coordinate system of $\mathbf{x}$ yields the same stochastic process $\mathbf{z}_\gamma$ up to a reparameterization of the Wiener process. Consequently, the distances introduced in the previous sections are invariant to Euclidean isometries as well.

**Proposition 2.** *Let $\phi(x) = Ax + b$ with $A \in \mathbb{R}^{d\times d}$ orthogonal ($A^\top A = I$) and $b \in \mathbb{R}^d$. Define*

$$\mathbf{y}_\gamma = \gamma\mathbf{x} + \mathbf{w}_\gamma,$$
$$\mathbf{y}_\gamma^\phi = \gamma\phi(\mathbf{x}) + \mathbf{w}_\gamma,$$

*where $\mathbf{w}_\gamma \sim \mathcal{N}(0, \gamma I)$ is a standard Wiener process statistically independent of $\mathbf{x}$. Then for all $x_1, x_2 \in \mathbb{R}^d$ and all $\gamma \geq 0$, we have*

$$\log\left(\frac{p_{\mathbf{y}_\gamma^\phi}(\gamma\phi(\mathbf{x}_2) + \mathbf{w}_\gamma)}{p_{\mathbf{y}_\gamma^\phi}(\gamma\phi(\mathbf{x}_1) + \mathbf{w}_\gamma)}\right) = \log\left(\frac{p_{\mathbf{y}_\gamma}(\gamma x_2 + \mathbf{w}'_\gamma)}{p_{\mathbf{y}_\gamma}(\gamma x_1 + \mathbf{w}'_\gamma)}\right),$$

*where $\mathbf{w}'_\gamma := A^\top\mathbf{w}_\gamma$ is also a standard Wiener process ($\mathbf{w}'_\gamma \overset{d}{=} \mathbf{w}_\gamma$).*

*Proof.* Let $\mathbf{w}'_\gamma := \boldsymbol{A}^\top \mathbf{w}_\gamma$. Since $\boldsymbol{A}$ is orthogonal, $\mathbf{w}'_\gamma \sim \mathcal{N}(\mathbf{0}, \gamma\boldsymbol{I})$ and remains a standard Wiener process independent of $\mathbf{x}$. Define $\mathbf{y}'_\gamma := \gamma\mathbf{x} + \mathbf{w}'_\gamma$. Then $\mathbf{y}'_\gamma \stackrel{d}{=} \mathbf{y}_\gamma$ and

$$
\begin{aligned}
\mathbf{y}^\phi_\gamma &= \gamma\boldsymbol{A}\mathbf{x} + \gamma\boldsymbol{b} + \mathbf{w}_\gamma \\
&= \boldsymbol{A}(\gamma\mathbf{x} + \boldsymbol{A}^\top\mathbf{w}_\gamma) + \gamma\boldsymbol{b} \\
&= \boldsymbol{A}\mathbf{y}'_\gamma + \gamma\boldsymbol{b}.
\end{aligned}
\tag{64}
$$

Now, since $|\det \boldsymbol{A}| = 1$, the change of variables formula gives, for every $\boldsymbol{y}$,

$$
\begin{aligned}
p_{\mathbf{y}^\phi_\gamma}(\boldsymbol{y}) &= p_{\boldsymbol{A}\mathbf{y}'_\gamma + \gamma\boldsymbol{b}}(\boldsymbol{y}) \\
&= p_{\mathbf{y}'_\gamma}\big(\boldsymbol{A}^\top(\boldsymbol{y} - \gamma\boldsymbol{b})\big) \\
&= p_{\mathbf{y}_\gamma}\big(\boldsymbol{A}^\top(\boldsymbol{y} - \gamma\boldsymbol{b})\big).
\end{aligned}
\tag{65}
$$

Thus, for any fixed $\boldsymbol{x}_i$, it holds that

$$
\begin{aligned}
p_{\mathbf{y}^\phi_\gamma}(\gamma\phi(\boldsymbol{x}_i) + \mathbf{w}_\gamma) &= p_{\mathbf{y}_\gamma}\big(\boldsymbol{A}^\top(\gamma\boldsymbol{A}\boldsymbol{x}_i + \gamma\boldsymbol{b} + \mathbf{w}_\gamma - \gamma\boldsymbol{b})\big) \\
&= p_{\mathbf{y}_\gamma}\big(\gamma\boldsymbol{x}_i + \boldsymbol{A}^\top\mathbf{w}_\gamma\big).
\end{aligned}
\tag{66}
$$

Hence

$$
\log\left(\frac{p_{\mathbf{y}^\phi_\gamma}(\gamma\phi(\boldsymbol{x}_2) + \mathbf{w}_\gamma)}{p_{\mathbf{y}^\phi_\gamma}(\gamma\phi(\boldsymbol{x}_1) + \mathbf{w}_\gamma)}\right) = \log\left(\frac{p_{\mathbf{y}_\gamma}(\gamma\boldsymbol{x}_2 + \boldsymbol{A}^\top\mathbf{w}_\gamma)}{p_{\mathbf{y}_\gamma}(\gamma\boldsymbol{x}_1 + \boldsymbol{A}^\top\mathbf{w}_\gamma)}\right) = \log\left(\frac{p_{\mathbf{y}_\gamma}(\gamma\boldsymbol{x}_2 + \mathbf{w}'_\gamma)}{p_{\mathbf{y}_\gamma}(\gamma\boldsymbol{x}_1 + \mathbf{w}'_\gamma)}\right).
\tag{67}
$$

$\square$

**Invariance to sufficient statistics.** Traditionally, perceptual distance functions are obtained by first defining a stochastic representation $\mathbf{y}$ of $\mathbf{x}$ through a likelihood function $p_{\mathbf{y}|\mathbf{x}}$, and then "pulling back" the information geometry of this conditional density to the signal space, as given by the Fisher information of the representation. One can make a qualitative analogy to the IEM distance, interpreting the Gaussian channel $\mathbf{y}_\gamma$ as the "representation" of $\boldsymbol{x}$. The process $\mathbf{z}_\gamma(\boldsymbol{x}_1, \boldsymbol{x}_2)$ then compares the signals $\boldsymbol{x}_1$ and $\boldsymbol{x}_2$ by estimating each from its respective representation, and measuring the discrepancy in the resulting estimation errors back in the original signal space. Although the "pull back" is now done through estimation quantities rather than information geometry, one may wonder whether they share similar properties. In particular, an important property of the Fisher information metric is its invariance under sufficient statistics of the representation. We show below that $\mathbf{z}_\gamma$, and thus the IEM, is also invariant under sufficient statistics of $\mathbf{y}_\gamma$.

**Proposition 3.** *Let $\mathbf{y}^\mathcal{T}_\gamma = \mathcal{T}(\mathbf{y}_\gamma, \gamma)$ be a sufficient statistic of $\mathbf{y}_\gamma$ with respect to $\mathbf{x}$ for every $\gamma$, namely $\mathbf{y}_\gamma \leftrightarrow \mathbf{y}^\mathcal{T}_\gamma \leftrightarrow \mathbf{x}$ is a Markov chain. Then*

$$
\log\left(\frac{p_{\mathbf{y}^\mathcal{T}_\gamma}(\mathcal{T}(\gamma\boldsymbol{x}_2 + \mathbf{w}_\gamma, \gamma))}{p_{\mathbf{y}^\mathcal{T}_\gamma}(\mathcal{T}(\gamma\boldsymbol{x}_1 + \mathbf{w}_\gamma, \gamma))}\right) = \log\left(\frac{p_{\mathbf{y}_\gamma}(\gamma\boldsymbol{x}_2 + \mathbf{w}_\gamma)}{p_{\mathbf{y}_\gamma}(\gamma\boldsymbol{x}_1 + \mathbf{w}_\gamma)}\right).
$$

*Proof.* From the law of total expectation, we have

$$
\mathbb{E}[\mathbf{x} \,|\, \mathbf{y}_\gamma] = \mathbb{E}\big[\mathbb{E}\big[\mathbf{x} \,\big|\, \mathbf{y}_\gamma, \mathbf{y}^\mathcal{T}_\gamma\big] \,\big|\, \mathbf{y}_\gamma\big].
\tag{68}
$$

Since $\mathbf{y}_\gamma \leftrightarrow \mathbf{y}^\mathcal{T}_\gamma \leftrightarrow \mathbf{x}$ is a Markov chain, then $p_{\mathbf{x}\,|\,\mathbf{y}^\mathcal{T}_\gamma, \mathbf{y}_\gamma} = p_{\mathbf{x}\,|\,\mathbf{y}^\mathcal{T}_\gamma}$. So we have

$$
\mathbb{E}\big[\mathbf{x} \,\big|\, \mathbf{y}^\mathcal{T}_\gamma, \mathbf{y}_\gamma\big] = \mathbb{E}\big[\mathbf{x} \,\big|\, \mathbf{y}^\mathcal{T}_\gamma\big].
\tag{69}
$$

Substituting Eq. (69) into Eq. (68), we get

$$
\mathbb{E}[\mathbf{x} \,|\, \mathbf{y}_\gamma] = \mathbb{E}\big[\mathbb{E}\big[\mathbf{x} \,\big|\, \mathbf{y}^\mathcal{T}_\gamma\big] \,\big|\, \mathbf{y}_\gamma\big].
\tag{70}
$$

Finally, since $\mathbf{y}^\mathcal{T}_\gamma$ is a function of $\mathbf{y}_\gamma$, then $\mathbb{E}\big[\mathbf{x} \,\big|\, \mathbf{y}^\mathcal{T}_\gamma\big]$ is a function of $\mathbf{y}_\gamma$ as well. We can therefore pull $\mathbb{E}\big[\mathbf{x} \,\big|\, \mathbf{y}^\mathcal{T}_\gamma\big]$ out of the expectation, and get

$$
\mathbb{E}[\mathbf{x} \,|\, \mathbf{y}_\gamma] = \mathbb{E}\big[\mathbf{x} \,\big|\, \mathbf{y}^\mathcal{T}_\gamma\big].
\tag{71}
$$

From here, it is easy to see that the processes

$$\mathbf{z}_\gamma = \log \left( \frac{p_{\mathbf{y}_\gamma}(\gamma \boldsymbol{x}_2 + \mathbf{w}_\gamma)}{p_{\mathbf{y}_\gamma}(\gamma \boldsymbol{x}_1 + \mathbf{w}_\gamma)} \right), \quad \text{and} \tag{72}$$

$$\mathbf{z}_\gamma^{\mathcal{T}} = \log \left( \frac{p_{\mathbf{y}_\gamma^{\mathcal{T}}}(\mathcal{T}(\gamma \boldsymbol{x}_2 + \mathbf{w}_\gamma))}{p_{\mathbf{y}_\gamma^{\mathcal{T}}}(\mathcal{T}(\gamma \boldsymbol{x}_1 + \mathbf{w}_\gamma))} \right) \tag{73}$$

are exactly the same, as they only depend on $\mathbb{E}[\mathbf{x} \mid \mathbf{y}_\gamma]$ and $\mathbb{E}[\mathbf{x} \mid \mathbf{y}_\gamma^{\mathcal{T}}]$, respectively. $\quad\square$

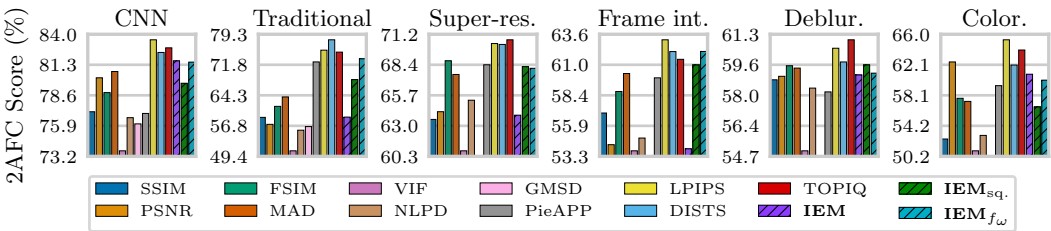

Figure 6: **Two-alternative forced choice (2AFC) performance comparison on the different distortion categories in the BAPPS dataset.** The unsupervised IEM$_{sq.}$ achieves competitive performance in most types of distortion. Our supervised variant, IEM$_{f_\omega}$, further improves the results.

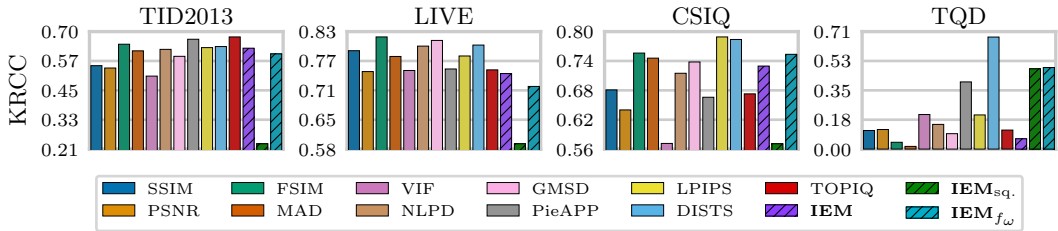

Figure 7: **Kendall correlation coefficient (KRCC) results on several full-reference image similarity benchmarks.**

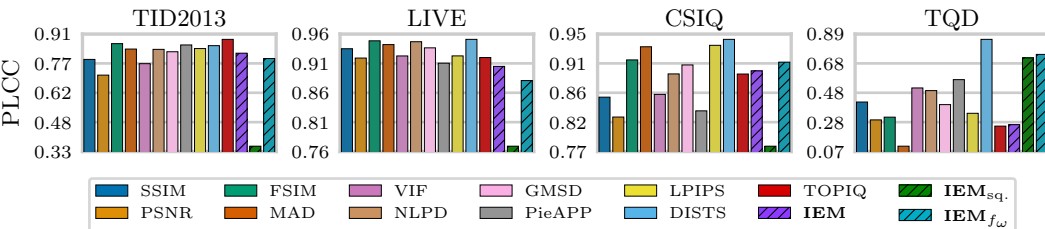

Figure 8: **Pearson linear correlation coefficient (PLCC) results on several full-reference image similarity benchmarks.** Following common practice, we map the similarity scores to the MOS scores by fitting a four-parameter logistic function.

# D ADDITIONAL EXPERIMENTAL RESULTS

## D.1 PREDICTING TWO-ALTERNATIVE FORCED CHOICE JUDGEMENTS

Here, we evaluate our solutions on the BAPPS dataset (Zhang et al., 2018), which consists of image triplets: a reference image $x_{ref}$ and two distorted versions, $x_1$ and $x_2$, along with a probability-of-preference score for each triplet. These probabilities are derived from human evaluations of similarity between the reference image and each distorted version. The probability score is defined as the fraction of evaluators who preferred $(x_{ref}, x_1)$ over $(x_{ref}, x_2)$. Specifically, if $n_1$ evaluators preferred $(x_{ref}, x_1)$ and $n_2$ preferred $(x_{ref}, x_2)$, then the triplet $(x_{ref}, x_1, x_2)$ is assigned the label $p = n_1/(n_1 + n_2)$. The two-alternative forced choice (2AFC) score for a given similarity measure is then computed by averaging $p \cdot \mathbf{1}_{s_1 < s_2} + (1-p) \cdot \mathbf{1}_{s_1 > s_2} + 0.5 \cdot \mathbf{1}_{s_1 = s_2}$ across the entire dataset (Zhang et al., 2018).

To compute the IEM, we train an additional neural denoiser model on ImageNet-1k with images of size $64 \times 64$, which is the native resolution of images in the BAPPS dataset. We use the same type of denoiser architecture as in Sec. 3 (see more details in App. E). Finally, we compute the IEM$_f$ using this trained denoiser, similarly to Sec. 3. The results are reported in Fig. 6.

### D.2 Predicting mean opinion scores: additional metrics

We extend the experiments from Sec. 3.2 by reporting additional metrics on MOS predictions for the same methods and datasets. In Figs. 7 and 8, we present Kendall's rank correlation coefficient (KRCC) and Pearson's linear correlation coefficient (PLCC), respectively. These correlation scores exhibit trends similar to those observed with Spearman's rank correlation coefficient (SRCC, see Fig. 4). Note that to compute the PLCC, we follow common practice and first fit a four-parameter logistic function.

### D.3 Maximum differentiation competition against PSNR: additional details and results

This section provides additional details and results for the maximum differentiation competition experiment described in Sec. 3.3.

**Overview of the maximum differentiation competition optimization procedure.** To conduct a maximum differentiation competition between a given perceptual distance measure and PSNR, we begin with a reference image $x$. We then sample a white Gaussian noise vector $\epsilon$ and normalize it such that the MSE between $x$ and $x + \epsilon$ equals a predetermined constant $C$, derived from a target PSNR level. Formally, this normalization is defined as

$$\epsilon \leftarrow C \frac{\epsilon}{\|\epsilon\|}, \tag{74}$$

where $C = 2\sqrt{d} \cdot 10^{-\text{PSNR}/20}$ and the image pixels value range is $[-1, 1]$. Given a total of $N$ optimization steps, we iteratively update $\epsilon$ by optimizing (minimizing or maximizing) the perceptual distance between $x$ and $x+\epsilon$ using projected gradient descent, where the radial projection in Eq. (74) is applied after each step to maintain a fixed PSNR throughout the optimization.

**Optimization of the IEM.** Using the IEM as an optimization objective is challenging (*e.g.*, when aiming to minimize the distance between a pair of images). In particular, since the IEM involves computing a one-dimensional integral of denoising errors over a wide range of noise levels, the corresponding denoiser gradients must be computed and aggregated across this range. This results in large deviations in both the scale and variance of the loss gradients across noise levels, as the scale and variance of the denoising errors at low-SNR regions are substantially larger than those at high-SNR regions, causing the low-SNR regions to dominate the optimization. Reweighting the integral through a change of variables does not fully resolve this problem, as it does not address the variance issue. To alleviate this issue, we propose an *annealing* strategy that enables stable optimization of the IEM in the context of the maximum differentiation competition against PSNR.

For an integration range $[\Gamma_0, \Gamma]$ in Def. 1 (where $\Gamma_0$ is a small value that replaces the lower bound of the integral), we optimize the IEM by progressively optimizing the integrand in Def. 1, starting at the lowest SNR $\Gamma_0$ and gradually increasing the SNR until it reaches $\Gamma$. Formally, we construct a sequence $\alpha_1, \alpha_2, \ldots, \alpha_N$ uniformly spaced between $\log \Gamma_0$ and $\log \Gamma$, and define $\gamma_i = \exp(\alpha_i)$. Then, at each optimization step $i = 1, 2, \ldots, N$, we update $\epsilon$ by minimizing or maximizing the objective

$$\gamma_i \big\| \nabla \log p_{\mathbf{y}_{\gamma_i}}(\gamma_i x + \mathbf{w}_{\gamma_i}) - \nabla \log p_{\mathbf{y}_{\gamma_i}}(\gamma_i(x + \epsilon) + \mathbf{w}_{\gamma_i}) \big\|^2, \tag{75}$$

where $\mathbf{w}_{\gamma_i} \sim \mathcal{N}(0, \gamma_i \mathbf{I})$ is sampled randomly and independently at each step $i$. Note that the optimization objective in Eq. (75) corresponds to the integrand in the definition of the IEM (Def. 1), where the expectation is approximated with a single noise sample (similarly to standard stochastic optimization), and the change of variables $\alpha = \log \gamma$ is applied (which introduces a multiplicative factor $\gamma$, as in Eq. (75)).

The distortion $\epsilon$ is optimized using the Adam optimizer (Kingma, 2014), whose momentum term aggregates gradients across SNR values over the $N$ optimization steps. Thus, the update at iteration $i$ implicitly incorporates information from all previous steps. After each update, we re-normalize $\epsilon$ according to Eq. (74) to ensure that the MSE between $x$ and $x + \epsilon$ remains equal to $C$ throughout the optimization. We fix $\Gamma_0 = 10^{-6}$ (lower bound of the integral) and $\Gamma = 1$ (upper bound of the integral) and use a learning rate of $5 \times 10^{-2}$ with $N = 1000$ optimization steps. The Adam running

averages coefficients are $\beta_1 = 0.9$ and $\beta_2 = 0.999$. These hyperparameters are kept fixed across all image examples.

**Optimization of other perceptual distance measures.** To illustrate the difference between the IEM and other perceptual distance measures (DISTS, LPIPS, VIF, SSIM, TOPIQ, PieAPP, NLPD, and GMSD), we use each measure to minimize or maximize the perceptual distance between $x$ and $x + \epsilon$, while maintaining a fixed PSNR level (we apply the same $\epsilon$ projection procedure after each optimization step, as defined in Eq. (74)). All optimizations are performed using the Adam optimizer with a learning rate of $5 \times 10^{-2}$, running averages coefficients of $\beta_1 = 0.9$ and $\beta_2 = 0.999$, and $N = 1000$ optimization steps (consistent with the optimization procedure used for the IEM). We find that all evaluated perceptual distance measures, including the IEM, are effectively optimized using these hyperparameter settings, as shown in Fig. 13. We employ the implementations of these metrics provided by the `IQA-PyTorch` package (Chen & Mo, 2022).

We note that TOPIQ, VIF, and SSIM are perceptual *similarity* measures, for which higher values indicate smaller perceptual distances. Thus, for these measures, minimizing (or maximizing) the perceptual distance corresponds to maximizing (or minimizing) the similarity measure.

**Results.** We conduct the maximum differentiation competition described above on several images selected from the DIV2K database (Agustsson & Timofte, 2017), which contains high-quality general-content images. Similarly to Sec. 3.2, we preprocess each image by first center-cropping it to the length of its shorter edge, and then resizing it to $256 \times 256$. For each method, the maximum differentiation competition is performed five times under different PSNR constraints of 30dB, 25dB, 20dB, 15dB, and 10dB. The final results are shown in Figs. 9 to 12.

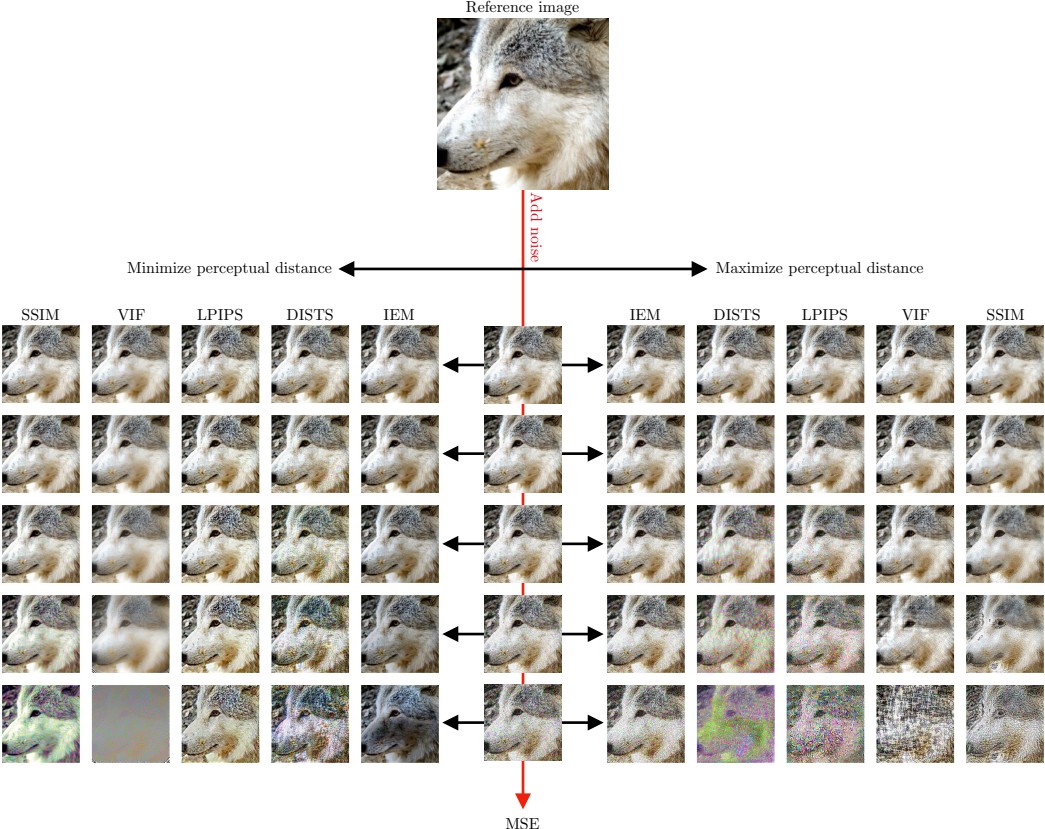

Figure 9: **Visual results of the maximum differentiation competition.** A reference image (top) is corrupted with increasing levels of white Gaussian noise (30dB, 25dB, 20dB, 15dB, and 10dB), producing a sequence of distorted images (middle column) with varying MSE (PSNR) values relative to the reference. Starting from each noise-distorted image, we minimize or maximize each of the perceptual distance measures shown (IEM, DISTS, LPIPS, VIF, and SSIM), while keeping the MSE (equivalently, PSNR) fixed. Interestingly, minimizing the IEM consistently yields high perceptual quality images that preserve the overall geometric structure of the reference image, even under large MSE (low PSNR) constraints. In contrast, all other methods introduce noticeable and unnatural artifacts during optimization (zoom in for best view). Maximizing the IEM reveals that it is most sensitive to unstructured noise perturbations that push an image off the "data support." This is consistent with our theoretical analysis in Sec. 2.

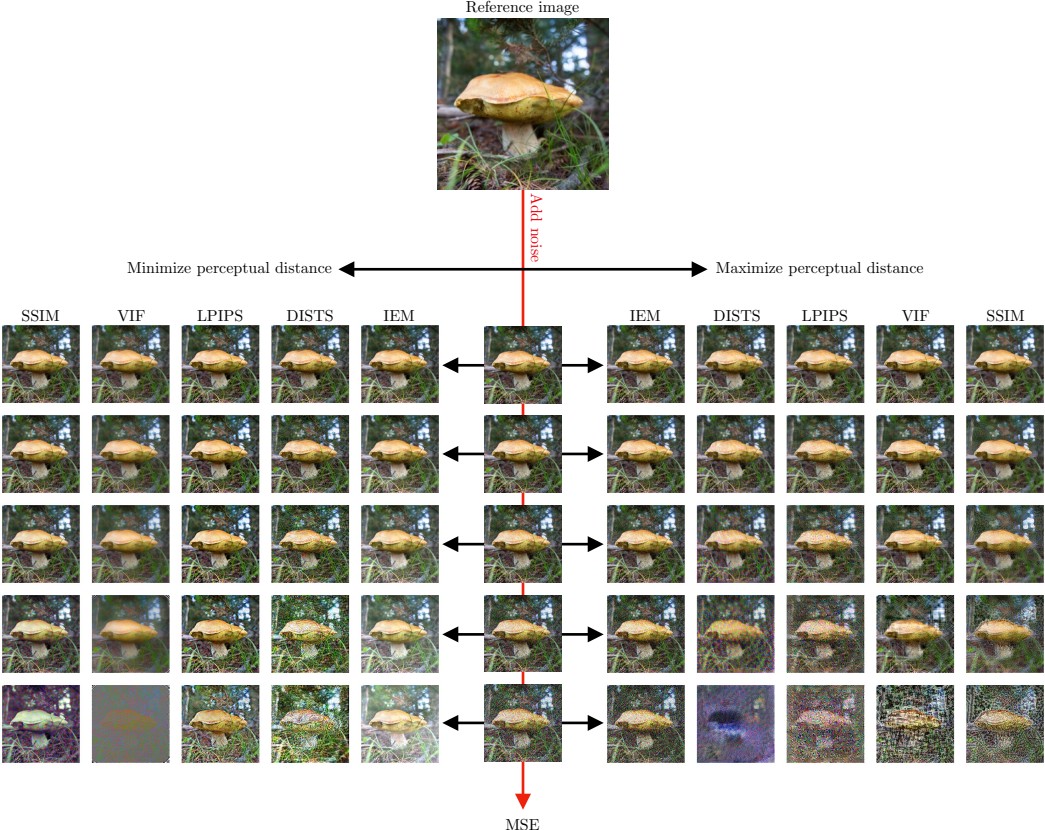

Figure 10: **Visual results of the maximum differentiation competition.** A reference image (top) is corrupted with increasing levels of white Gaussian noise (30dB, 25dB, 20dB, 15dB, and 10dB), producing a sequence of distorted images (middle column) with varying MSE (PSNR) values relative to the reference. Starting from each noise-distorted image, we minimize or maximize each of the perceptual distance measures shown (IEM, DISTS, LPIPS, VIF, and SSIM), while keeping the MSE (equivalently, PSNR) fixed. Interestingly, minimizing the IEM consistently yields high perceptual quality images that preserve the overall geometric structure of the reference image, even under large MSE (low PSNR) constraints. In contrast, all other methods introduce noticeable and unnatural artifacts during optimization (zoom in for best view). Maximizing the IEM reveals that it is most sensitive to unstructured noise perturbations that push an image off the "data support." This is consistent with our theoretical analysis in Sec. 2.

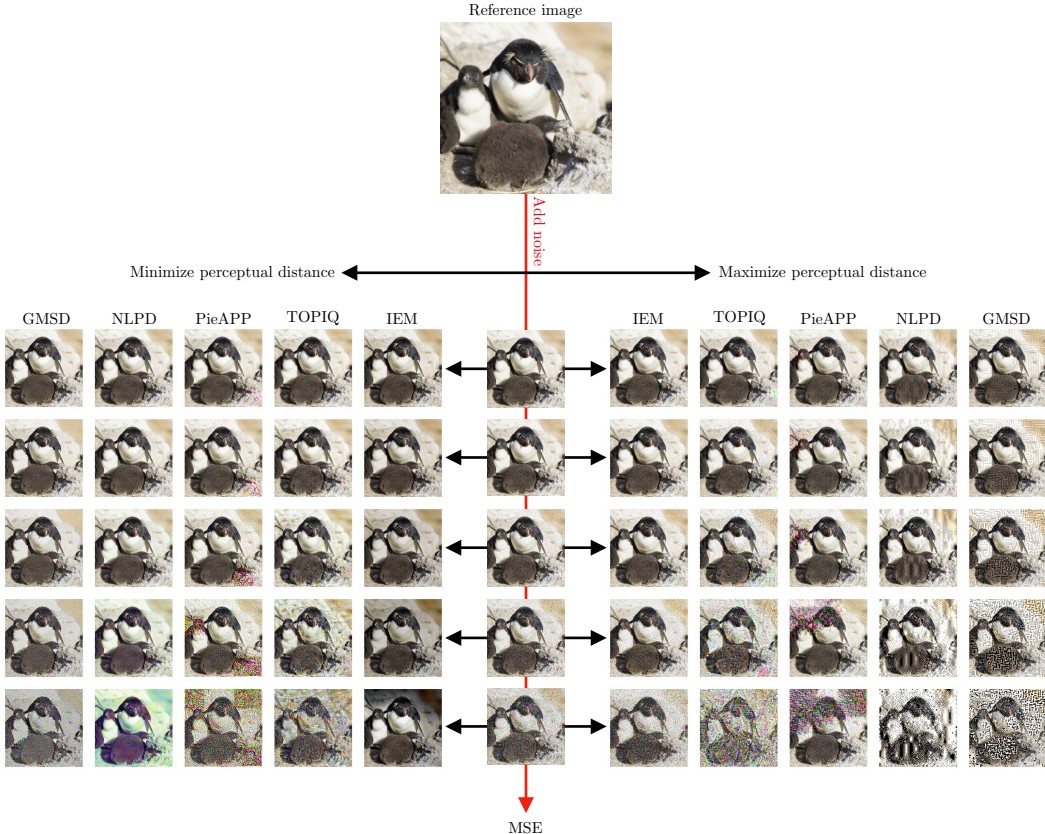

Figure 11: **Visual results of the maximum differentiation competition.** A reference image (top) is corrupted with increasing levels of white Gaussian noise (30dB, 25dB, 20dB, 15dB, and 10dB), producing a sequence of distorted images (middle column) with varying MSE (PSNR) values relative to the reference. Starting from each noise-distorted image, we minimize or maximize each of the perceptual distance measures shown (IEM, TOPIQ, PieAPP, NLPD, and GMSD), while keeping the MSE (equivalently, PSNR) fixed. Interestingly, minimizing the IEM consistently yields high perceptual quality images that preserve the overall geometric structure of the reference image, even under large MSE (low PSNR) constraints. In contrast, all other methods introduce noticeable and unnatural artifacts during optimization (zoom in for best view). Maximizing the IEM reveals that it is most sensitive to unstructured noise perturbations that push an image off the "data support." This is consistent with our theoretical analysis in Sec. 2.

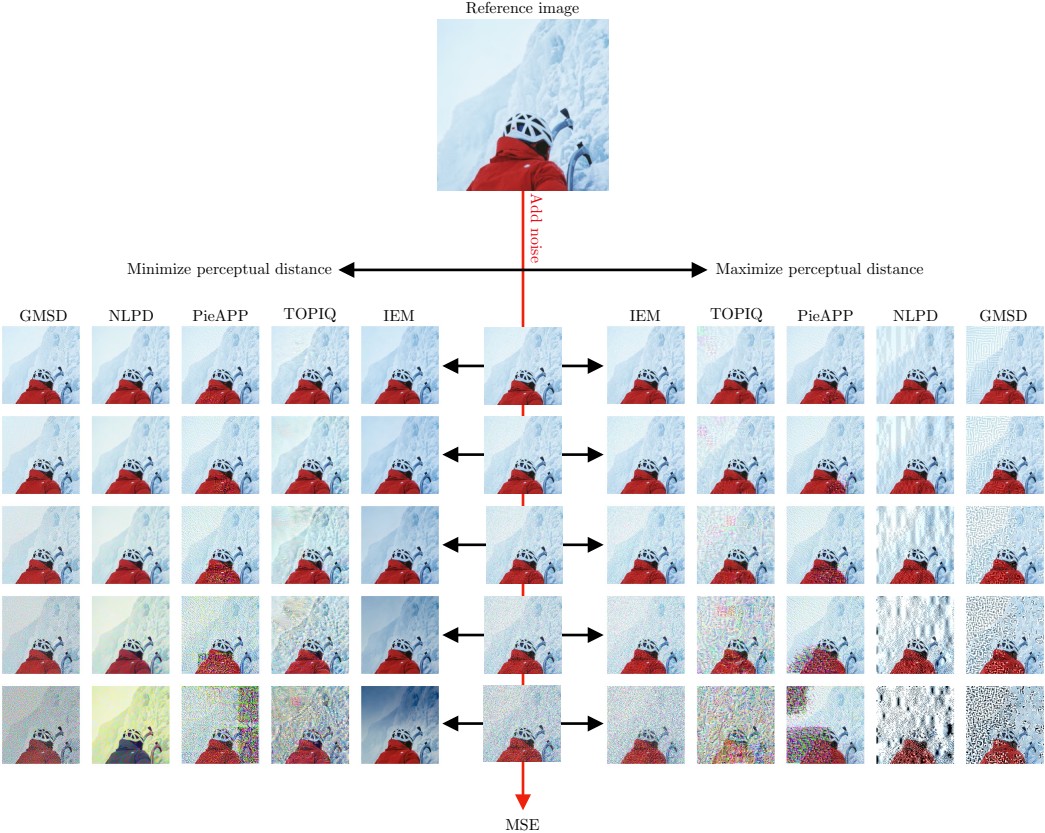

Figure 12: **Visual results of the maximum differentiation competition.** A reference image (top) is corrupted with increasing levels of white Gaussian noise (30dB, 25dB, 20dB, 15dB, and 10dB), producing a sequence of distorted images (middle column) with varying MSE (PSNR) values relative to the reference. Starting from each noise-distorted image, we minimize or maximize each of the perceptual distance measures shown (IEM, TOPIQ, PieAPP, NLPD, and GMSD), while keeping the MSE (equivalently, PSNR) fixed. Interestingly, minimizing the IEM consistently yields high perceptual quality images that preserve the overall geometric structure of the reference image, even under large MSE (low PSNR) constraints. In contrast, all other methods introduce noticeable and unnatural artifacts during optimization (zoom in for best view). Maximizing the IEM reveals that it is most sensitive to unstructured noise perturbations that push an image off the "data support." This is consistent with our theoretical analysis in Sec. 2.

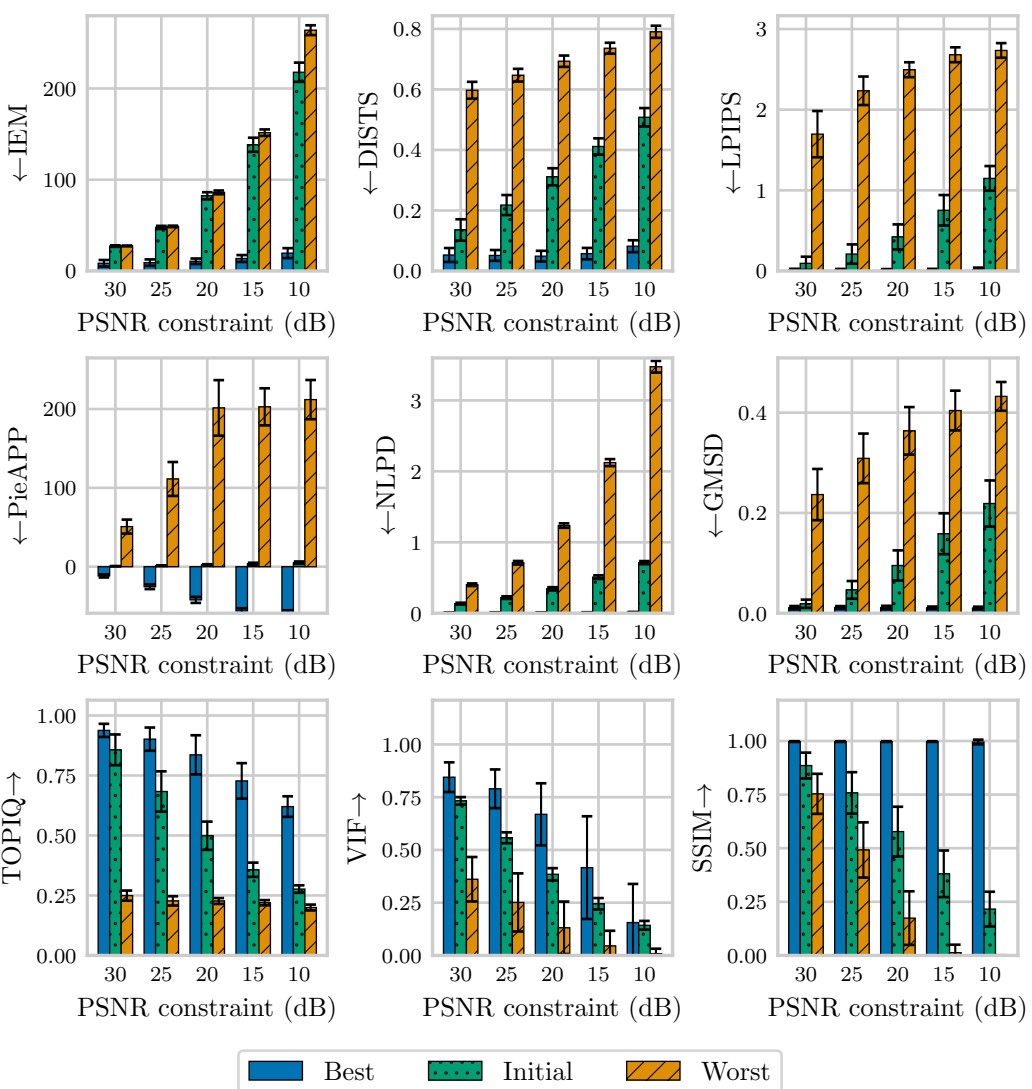

Figure 13: **Best, worst, and initial values of different perceptual metrics in the maximum differentiation competition against PSNR.** To verify that the evaluated perceptual metrics are effectively optimized in the maximum differentiation competition against PSNR, we present the average best, worst, and initial values of each metric (shown as bar plots), together with their standard deviations (shown as error bars), computed over 30 images randomly selected from the DIV2K dataset. The best and worst values correspond to the final perceptual distances obtained through minimization or maximization of the distance under a fixed PSNR constraint, respectively, while the initial value represents the perceptual distance between the initial distorted (noisy) image and the reference image. While global optima are not guaranteed, these results show that all perceptual distance measures are consistently optimized. For example, the best SSIM values remain close to their maximum possible value of 1 across all PSNR constraints, whereas the worst SSIM values approach 0 as we decrease the PSNR constraint. This trend is consistent across all evaluated perceptual metrics. Thus, for each perceptual metric, the images on the left-hand side of Figs. 9 to 12 (in the column corresponding to that metric) are perceived as similar to the reference image according to the metric, whereas the images on the right-hand side are perceived as perceptually distant from the reference image. Finally, we note that TOPIQ, VIF, and SSIM are perceptual *similarity* measures, for which higher values indicate smaller perceptual distances. Thus, for these measures, minimizing (or maximizing) the perceptual distance corresponds to maximizing (or minimizing) the measure. For this reason, the "best" (blue) bar plots for these metrics are high, while the "worst" (orange) bar plots are low.

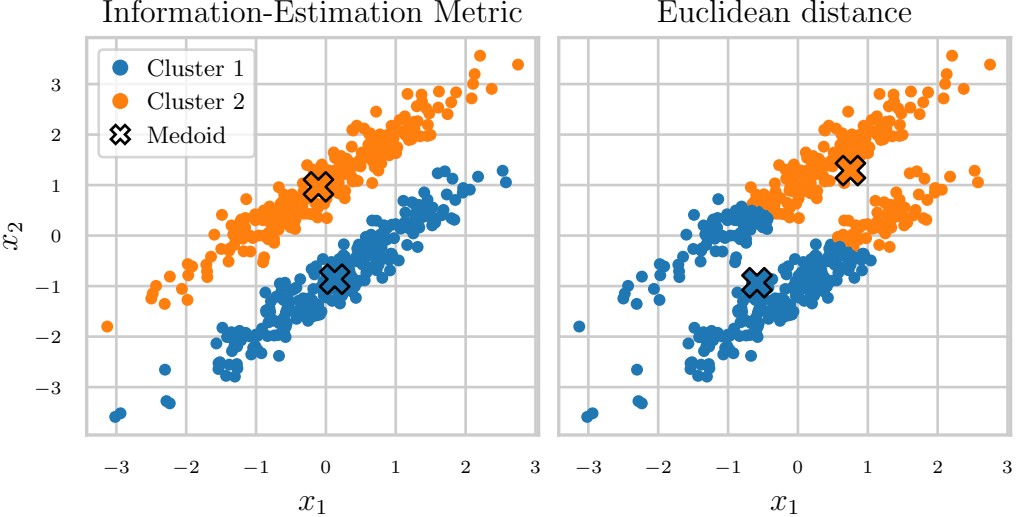

Figure 14: **Utilizing the IEM to solve a toy clustering Gaussian mixture problem.** We apply the K-medoids algorithm twice: once with the IEM (left panel) and once with the Euclidean distance (right panel). While K-medoids with the Euclidean distance fails to recover the correct cluster separation, using the IEM yields an accurate solution that aligns with the underlying modes.

### D.4 A TOY CLUSTERING EXAMPLE

To illustrate a potential future application of the IEM, we use it to solve a simple two-dimensional clustering problem in which the K-medoids algorithm (Kaufman & Rousseeuw, 2008), when coupled with the Euclidean distance, fails to provide a satisfactory result. Specifically, we consider a Gaussian mixture density consisting of two modes,

$$p_{\mathbf{x}}(\boldsymbol{x}) = 0.5\mathcal{N}\left(\boldsymbol{x}; \begin{pmatrix} 0 \\ 1 \end{pmatrix}, \begin{pmatrix} 1 & 0.95 \\ 0.95 & 1 \end{pmatrix}\right) + 0.5\mathcal{N}\left(\boldsymbol{x}; \begin{pmatrix} 0 \\ -1 \end{pmatrix}, \begin{pmatrix} 1 & 0.95 \\ 0.95 & 1 \end{pmatrix}\right). \tag{76}$$

We apply the K-medoids algorithm on 500 samples drawn independently from this density, using either the IEM or the Euclidean distance as the distance measure. To compute the IEM, we train a simple unsupervised neural denoiser (a 5-layer fully-connected network with GELU activations) on the range $\log \gamma \in [2^{-10}, 2^{10}]$, and compute the integral in Def. 1 on the same range after changing the integration variable to $\log \gamma$. Importantly, the denoiser only depends on $\mathbf{y}_\gamma$ and $\gamma$, so it is "unaware" of the clusters. We use 200 discretization steps to numerically solve the integral, and approximate the expectation of the integrand by averaging over 50 random Brownian motion paths. Remarkably, the IEM resolves the clustering problem effectively: it selects medoids located at the means of each mode and assigns samples to their corresponding modes with high accuracy, as illustrated in Fig. 14.

## E TRAINING AND IMPLEMENTATION DETAILS

### E.1 NUMERICAL INTEGRATION

Computing any of our distance functions requires numerically solving an SDE. We use the Euler–Maruyama discretization for this purpose, applying a change of variables so that the integral is evaluated over log-SNR values instead of SNR values. This common technique improves numerical stability and distributes integration steps more effectively across the SNR domain. To reduce computational demands, we compute the integrals along a single Brownian motion path in all our experiments. Compared with averaging over multiple paths, we observe no significant difference in the resulting approximated distances. An independent Brownian motion path is generated randomly each time we compute the distance. Throughout our experiments, we use 512 discretization steps, which is a relatively large number. This choice ensures that our results reflect the intended distance

measures more accurately. Empirically, we find that using fewer steps (*e.g.*, 128) does not alter the results.

### E.2 DENOISER TRAINING

The HDiT denoiser model training hyper-parameters are given in Tab. 1. To resize the ImageNet-1k training images to $256 \times 256$ resolution, we center-crop each image to its shorter edge dimension and then use Lanczos resampling. For ImageNet-1k $64 \times 64$, we use the official resized training set provided on the ImageNet website.

We note that our trained denoiser model uses the Variance Exploding (VE) formulation (Song et al., 2020) (specifically, EDM (Karras et al., 2022)), where the SNR $\gamma$ and the noise level $\sigma$ are related via $\gamma = 1/\sigma^2$.

### E.3 LEARNING $f'_\omega$

We describe the architecture of our learned $f'_\omega$ in Tab. 2. This function contains about 3M parameters in all our experiments.

In Sec. 3.2, we train $f'_\omega$ using the KADID-10k (Lin et al., 2019) and DTD (Cimpoi et al., 2014) datasets, similarly to (Ding et al., 2022). We cap the integral in Def. 2 at $\Gamma = 10^2$. For KADID-10k, we employ a simple pairwise ranking loss to encourage agreement between the rankings of the predicted distances and the ground-truth MOS scores. Specifically, given a mini-batch of triplets

$$\{(\boldsymbol{x}^{(i)}_{\text{ref.}}, \boldsymbol{x}^{(i)}_{\text{dist.}}, s^{(i)})\}^N_{i=1}, \tag{77}$$

where $\boldsymbol{x}^{(i)}_{\text{ref.}}$ is a reference image, $\boldsymbol{x}^{(i)}_{\text{dist.}}$ is its distorted version, and $s^{(i)}$ is the MOS score for this pair, we compute the distances

$$d^{(i)} = \text{IEM}_{f_\omega}\left(\boldsymbol{x}^{(i)}_{\text{ref.}}, \boldsymbol{x}^{(i)}_{\text{dist.}}\right). \tag{78}$$

The pairwise logistic ranking loss is then defined as

$$\mathcal{L} = \frac{1}{N^2} \sum_{i,j} \log\left(1 + \exp\left(-\tfrac{1}{\tau}(s^{(i)} - s^{(j)})(d^{(i)} - d^{(j)})\right)\right), \tag{79}$$

where $\tau$ is a temperature parameter learned jointly with $\omega$. For DTD, we randomly crop subimages of size $256 \times 256$ from each texture image, and apply a smooth $L_1$ loss with $\beta = 10.0$ to the distances obtained for these patches, encouraging $\text{IEM}_{f_\omega}$ to be small for textures of the same kind.

In App. D.1, we train $f'_\omega$ using the training split of BAPPS. We cap the integral in Def. 2 at $\Gamma = 10^6$. Similarly to Zhang et al. (2018), the loss function is a standard cross-entropy loss applied to the output of a small fully connected network with nonlinear activations. This network maps the raw distances $\text{IEM}_{f_\omega}$ to logits, which are then passed to the cross-entropy loss. The additional fully connected network is trained jointly with $f'_\omega$.

In all experiments, we use the Adam optimizer (Kingma, 2014) with a learning rate of $10^{-3}$, a batch size of 512, and train for 100 epochs. We apply an exponential learning rate decay with a factor 0.95 after each epoch.

### E.4 IMPLEMENTATION OF THE ILLUSTRATIVE EXAMPLES

We provide additional illustrative examples in Fig. 15, where we consider three different one-dimensional prior densities.

**Prior densities used in Fig. 2.** In the middle column of Fig. 2, we use a Gaussian density

$$p_{\mathbf{x}}(\boldsymbol{x}) = \mathcal{N}\left(\boldsymbol{x}; \begin{pmatrix} 0 \\ 1 \end{pmatrix}, \begin{pmatrix} 1 & 0 \\ 0 & 0.1 \end{pmatrix}\right). \tag{80}$$

In the right-most column of Fig. 2, we use a Gaussian mixture density

$$p_{\mathbf{x}}(\boldsymbol{x}) = 0.3\mathcal{N}\left(\boldsymbol{x}; \begin{pmatrix} 0 \\ 1 \end{pmatrix}, \begin{pmatrix} 1 & 0 \\ 0 & 0.1 \end{pmatrix}\right) + 0.7\mathcal{N}\left(\boldsymbol{x}; \begin{pmatrix} 1 \\ -1 \end{pmatrix}, \begin{pmatrix} 1 & 0.5 \\ 0.5 & 0.4 \end{pmatrix}\right). \tag{81}$$

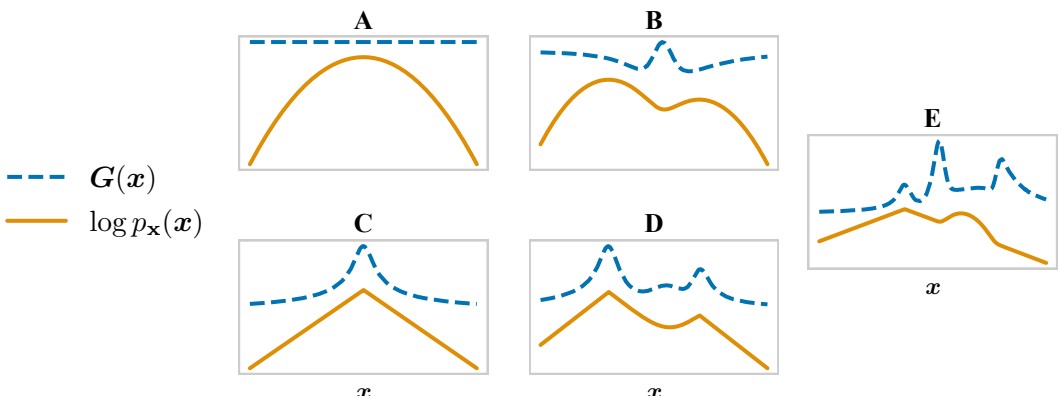

Figure 15: **Local sensitivity of the Information–Estimation Metric (IEM) for several one-dimensional prior densities.** In each subplot, the local metric $G(x)$ (Eq. (5), using a very large $\Gamma$) is plotted and compared to the log-density $\log p_{\mathbf{x}}$. The integral in Eq. (5) is computed over $\log \gamma \in [2^{-5}, 2^5]$ after applying change of variables. **(A)** For a Gaussian (light-tailed) density, $G(x)$ coincides with the Mahalanobis metric, which is constant everywhere. **(B)** For a Gaussian mixture density, $G(x)$ decreases near the local maxima and increases near the local minima; it converges to a constant value only at the tails. **(C)** For a Laplace (heavy-tailed) density, the sensitivity of the local metric increases with the probability of the signal $x$. **(D)** For a Laplace mixture density, the sensitivity grows near both the local maxima and minima, where the absolute curvature of the log-density is relatively large. **(E)** For a mixture of Laplace and Gaussian densities (left and right modes, respectively), the sensitivity is relatively constant only in the region where the Gaussian density dominates the Laplace density. All of these plots illustrate that the sensitivity of $G(x)$ is governed by the local curvature of $\log p_{\mathbf{x}}$ around $x$.

In the left-most column of Fig. 2, we use a two-dimensional Laplace density obtained by taking the product of two one-dimensional densities,

$$p_{\mathbf{x}}(\boldsymbol{x}) = \text{Lap}(x_1; \mu = 0, \, b = 0.3) \times \text{Lap}(x_2; \mu = 1, \, b = 0.1), \qquad (82)$$

where $\boldsymbol{x} = (x_1, x_2)$.

**Numerical computation of the IEM and the associated local metric $G(\boldsymbol{x}, \Gamma)$.** For each of the prior densities above, we write the function $\log p_{\mathbf{y}_\gamma}$ in closed form in PyTorch and compute $\nabla \log p_{\mathbf{y}_\gamma}$ and $\nabla^2 \log p_{\mathbf{y}_\gamma}$ using `torch.autograd`. The IEM in Def. 1 (global distance) is computed using 200 uniformly spaced discretization steps over $\log \gamma \in [2^{-10}, 2^{10}]$ for all prior densities, with 50 random Brownian-motion paths to estimate expectations. The local metric $G(\boldsymbol{x}, \Gamma)$ in Eq. (5) is computed using the same number of discretization steps and Brownian-motion paths, while the integral is taken over $\log \gamma \in [2^{-4}, 2^4]$.

## F  LLMS USAGE

We used Large Language Models (LLMs) for minor text polishing and assistance in generating figures.

Table 1: HDiT architecture details and training hyperparameters for our two configurations.

| Hyper-parameter | ImageNet-64$^2$ | ImageNet-256$^2$ |
|---|---|---|
| Parameters | 11.6M | 22.1M |
| Training steps | 400k | 400k |
| Batch size | 256 | 256 |
| Image size | 64×64 | 256×256 |
| Precision | bfloat16 | bfloat16 |
| Training hardware | 1 H100 80GB | 1 H100 80GB |
| Training time | 1 day | 3 days |
| Patch size | 4 | 4 |
| Levels (local + global attention) | 1 + 1 | 1 + 1 |
| Depth | [2, 11] | [2, 11] |
| Widths | [64, 128] | [128, 256] |
| Attention heads (width / head dim.) | [1, 2] | [2, 4] |
| Attention head dim. | 64 | 64 |
| Neighborhood kernel size | 7 | 7 |
| Mapping depth | 1 | 1 |
| Mapping width | 768 | 768 |
| Data sigma | 0.5 | 0.5 |
| Sigma sampling density | log-uniform | log-uniform |
| Sigma range | $[10^{-3}, 10^3]$ | $[10^{-3}, 10^3]$ |
| Optimizer | AdamW | AdamW |
| Learning rate | $5 \cdot 10^{-4}$ | $5 \cdot 10^{-4}$ |
| Learning rate scheduler | Constant (no warmup) | Constant (no warmup) |
| AdamW betas | [0.9, 0.95] | [0.9, 0.95] |
| AdamW eps. | $10^{-8}$ | $10^{-8}$ |
| Weight decay | $10^{-2}$ | $10^{-2}$ |
| EMA decay | 0.9999 | 0.9999 |

Table 2: Architecture and hyperparameters of our $f_\omega$ network.

| Layer | Details | Output Shape |
|:---:|:---:|:---:|
| Input | $\{z_{\gamma_i}\}_{i=1}^{512} \in \mathbb{R}^{512}$ | 512 |
| Absolute value | $z_{\gamma_i} \leftarrow |z_{\gamma_i}|$ | 512 |
| Scale | Multiply by learnable $\alpha \in \mathbb{R}^{512}$ | 512 |
| 1–5 | MaskedLinear(512, 512)[1]
+ GELU
+ Dropout(0.1) | 512 |
| Output | MaskedLinear(512, 512)[1] | 512 |
| Final transform | $\log(1 + x^2)$ | 512 |

[1] **MaskedLinear:** a fully connected linear layer with a lower-triangular binary mask applied to its weight matrix. This enforces a causal (autoregressive) structure by ensuring that the $i$-th output depends only on the first $i$ inputs.

