# OpenReview forum: "Learning a distance measure from the information-estimation geometry of data"
_ICLR.cc/2026/Conference — ICLR 2026 Poster_

### Official Review · Reviewer_dR9Y · 2025-10-26

**Soundness:** 3
**Presentation:** 3
**Contribution:** 3
**Rating:** 6
**Confidence:** 3

**Summary:**

This paper innovatively proposes the Information-Estimation Metric (IEM), a distance measure derived from the local geometric properties of probability distributions (denoising errors and score vector fields). It breaks through the limitation that traditional supervised perceptual metrics rely on manual annotations, validates its unsupervised learning capability on the ImageNet dataset, and outperforms most existing methods in image quality assessment tasks. The overall theoretical derivation is rigorous and the experimental design is comprehensive.

**Strengths:**

1.Solid Mathematical Foundation. The paper reliably demonstrates the mathematical significance of its work.
2.Comprehensive Experimental Validation and Credible Conclusions. The paper verifies the advantages of IEM over other metrics through experiments and tests.

**Weaknesses:**

1. The drawback of low computational efficiency of the IEM.
2.The experimental scope is relatively limited, with insufficient validation of the IEM’s performance across a wide range of noise types.

**Questions:**

More experiments on different datasets should be more convincing.
Could you provide more evidence of the demand for IEM?

---

> ### Author Response · Authors · 2025-11-20
>
> We thank the reviewer for their positive and encouraging review, and for recognizing both the theoretical rigor and the experimental comprehensiveness of our work. We address the concerns raised below.
>
> ## Computational efficiency
> The IEM is more computationally demanding than standard perceptual metrics, which require a single network evaluation. This is because the IEM involves an integral over denoising errors, whose numerical computation requires multiple evaluations of a denoiser (analogous to the fact that diffusion models typically require multiple denoiser evaluations to generate samples). While we mention this as an explicit limitation in our paper, we also view it as an opportunity for future work—much like the early versions of diffusion models, which required hundreds of denoising steps but later became far more efficient through improved formulations. We are hopeful that the IEM will see similar progress. For instance, we are currently exploring a formulation that replaces numerical integration with a single-pass evaluation—much like single-step diffusion sampling methods.
>
> ## Broader experimental validation
> We appreciate the reviewer’s suggestion to test the IEM on a wider range of noise and distortion types. Our current evaluation already encompasses five large-scale and widely used human-annotated perceptual quality databases—LIVE, TID2013, CSIQ, TQD, and BAPPS—which together cover over 25 distinct distortion categories, including Gaussian noise, blur, JPEG compression, color shifts, and texture variations. These datasets serve as standard benchmarks in research on perceptual distance measures, and encompass a broad range of distortions observed in image processing tasks. We are not aware of IQA databases that are categorically different or more comprehensive in terms of the types of distortion they cover.
>
> Furthermore, we have now included additional experiments in Sec. 3.3 and App. D.3, where we minimize or maximize the IEM or other perceptual metrics under a fixed PSNR constraint. Interestingly, even with a very low PSNR constraint of 10dB, we find that minimizing the IEM yields artifact-free images that are perceptually similar to the reference image, whereas minimizing state-of-the-art supervised perceptual metrics, such as LPIPS or DISTS, yields unrealistic images with noticeable artifacts. This suggests that the IEM is a robust distance function—not only capable of detecting a limited set of perceptually noticeable distortions (as shown in Sec. 3) but also able to anticipate perceptually noticeable distortions that may arise during optimization.
> We agree that exploring additional domains would further illustrate the generality of our approach. However, it is beyond the scope of our paper, and we view it as a promising option for future work.
>
> ## Broader motivation and demand for the IEM
> We appreciate the opportunity to clarify the motivation behind the IEM. Our work addresses a long-standing challenge: deriving an unsupervised, mathematically interpretable, and computationally tractable distance measure that aligns well with human perceptual judgments of image similarities. As discussed in the introduction, such a distance is highly desirable both from a scientific perspective—providing insight into the relationship between perception and natural signal statistics—and from a practical perspective: image processing algorithms (e.g., compression, restoration, or synthesis methods) should ideally be evaluated in terms of human perception.
> Existing state-of-the-art perceptual metrics (e.g., LPIPS, DISTS) achieve strong performance but rely on supervised training from labeled datasets involving human judgments. This dependence limits their applicability to domains where such labels are available and constrains their theoretical interpretability. In contrast, the IEM is defined directly in terms of the image data distribution itself, eliminating the need for annotations or task-specific supervision. This also makes the IEM mathematically interpretable, as discussed in detail in Sec. 2. More broadly, the IEM represents a new form of distance function that is adapted to the shape of the data support. This property makes it a potentially valuable tool beyond predicting human perceptual judgements—for example, in dimensionality reduction, manifold learning, and other domains where understanding the geometry of continuous data is important.

---

> > ### Comment · Reviewer_dR9Y · 2025-11-26
> >
> > Thank you for the authors' response.
> > Regarding the motivation and necessity of the IEM: The authors have outlined the research motivation for the IEM. It is recommended that the authors supplement relevant content in the main text and provide a more detailed elaboration.
> > Regarding the experiments: The authors mentioned adding comparative experiments where the IEM is compared with other metrics as optimization objectives. This is essential for demonstrating the application value of the IEM.
> > Regarding the explanation of the IEM's computational efficiency: Compared with the contributions and validations of this work, the limitation of insufficient computational efficiency is indeed acceptable. We wish the subsequent related work every success.

---

> > > ### Author Response · Authors · 2025-11-28
> > > **Thank you!**
> > >
> > > We are grateful to the reviewer for addressing our rebuttal.
> > >
> > > 1. __Regarding the motivation__: We feel that we have already outlined the motivation for the IEM in the introduction and discussion sections, and we do not think the rebuttal contains any new information that is not already covered in the paper. We would appreciate more specific suggestions on how to improve the motivation in the main text.
> > > 2. __Regarding the new experiments__: We would like to emphasize that the new experiments are already included in Sec. 3.3 of the revised manuscript (which we have uploaded to OpenReview). We would be glad to receive the reviewer's feedback.
> > > 3. __Regarding the computational complexity__: Please note that the computational complexity is already mentioned as a limitation in the discussion section, in both the original and revised manuscripts.

---

### Official Review · Reviewer_CAxo · 2025-10-31

**Soundness:** 2
**Presentation:** 3
**Contribution:** 3
**Rating:** 6
**Confidence:** 2

**Summary:**

This paper proposes the IEM and primarily validates it on image quality assessment tasks. The core idea involves measuring distribution shifts, and the method is not inherently limited to visual data. The experimental results demonstrate its performance on specific datasets like TQD.

**Strengths:**

*   **General Formulation:** The proposed IEM is not fundamentally tied to visual data, suggesting potential applicability to other media types like audio or natural language, as it makes no specific assumptions about the modality.
*   **Interesting Motivation:** The motivation based on "humans exhibit complex patterns of sensitivity that vary with the direction of the signal’s perturbation" is compelling and aligns with nuanced aspects of perception.

**Weaknesses:**

*   **Insufficient Experimental Validation:**
    *   The paper's core motivation and problem formulation do not appear exclusively focused on Image Quality Assessment (IQA), yet validation is conducted *only* within IQA tasks. This is insufficient. If the goal is a general measure of distribution shift, experiments on other modalities (e.g., sound, text) are necessary to substantiate the claim of generality.
    *   Within the IQA validation, there is a lack of experiments on real-world images with complex, mixed distortions. This is crucial for verifying the method's ability to assess information changes caused by complicated, realistic impairments.
*   **Lack of Comparative Analysis:** A critical weakness is the absence of a comparative analysis between IEM and other established information-theoretic distance measures like Kullback-Leibler Divergence (KLD), Jensen-Shannon Divergence (JSD), and Earth Mover's Distance (EMD). Such a comparison is essential for demonstrating IEM's potential superiority or unique advantages.
*   **Inadequate Explanations:**
    *   The significant performance discrepancy of `IEM_sq` (poor on photo quality assessment but good on TQD) lacks a thorough explanation from either a theoretical or experimental analysis perspective.
    *   The notation `IEM_fw` is introduced without a clear explanation.
*   **Parameter Sensitivity:** The IEM demonstrates high sensitivity to the parameter `Γ` (Gamma), with the "optimal" value varying drastically (e.g., from `1/4` to `10^6`) between different IQA tasks (photographic vs. texture images). This instability is a practical concern that needs addressing.

**Questions:**

1.  **Generalizability and Scope:** The paper's motivation seems broader than IQA. Have you considered, or could you perform, validation on non-visual modalities (e.g., audio, text) to truly demonstrate IEM's general applicability for measuring distribution shifts?
2.  **Theoretical Interpretation and Measurement:** A key motivation relates to sensitivity varying with the *direction* of perturbation. In high-dimensional spaces (e.g., latent spaces), how can the "direction" of distributional change be effectively measured? Furthermore, how might the similarity or difference between these "directions" of change impact IEM's performance? Is IEM inherently sensitive to this directional information?
3.  **Application to Generative Models:** Could IEM be applied to quantify the difference between generated and real images, thus serving as a metric for evaluating the performance of generative models?
4.  **Comparative Advantage:** How does IEM's performance compare directly to other information-theoretic measures like KLD, JSD, and EMD on your tasks? This comparison is vital for establishing IEM's value.
5.  **Explanation of Results and Definitions:** What explains the contradictory performance of `IEM_sq` (good on TQD, poor on photo quality assessment)? Can you provide a theoretical hypothesis or detailed experimental analysis?
6.  **Robustness to Parameters:** The performance is highly sensitive to `Γ`. What is the underlying reason for this sensitivity? Are there strategies to make the method more robust to the choice of `Γ`, or to determine an optimal `Γ` in a more principled, task-agnostic way?
7.  **Sensitivity to Denoiser Choice:** Is the performance of the proposed method sensitive to the selection of the specific denoiser used?

---

> ### Author Response · Authors · 2025-11-19
>
> We thank the reviewer for the time and effort spent reviewing our work and for the interesting comments and questions.
> We address the weaknesses and questions below.
>
> ## Weakness 1: Insufficient Experimental Validation
> We would like to clarify that our current evaluation encompasses five large-scale and widely used human-annotated perceptual quality databases—LIVE, TID2013, CSIQ, TQD, and BAPPS—which together cover over 25 distinct distortion categories, including Gaussian noise, blur, JPEG compression, color shifts, and texture variations. These datasets serve as standard benchmarks in research on perceptual distance measures, since they encompass a broad range of distortions observed in natural image processing tasks.
>
> Furthermore, we have now included additional experiments in Sec. 3.3 and App. D.3, in which we minimize or maximize the IEM (and eight other perceptual metrics) under a fixed PSNR constraint. Interestingly, even with a very low PSNR constraint of 10dB, we find that minimizing the IEM yields artifact-free images that are perceptually similar to the reference image, whereas minimizing state-of-the-art supervised perceptual metrics, such as LPIPS or DISTS, yields unrealistic images with noticeable artifacts. This suggests that the IEM is a robust distance function that generalizes well over the full space of images—not only capable of detecting the specific distortions provided in the aforementioned databases (as shown in Sec. 3) but also able to anticipate perceptually noticeable distortions that may arise during optimization.
>
> ## Question 1: Have you considered, or could you perform, validation on non-visual modalities (e.g., audio, text) to truly demonstrate IEM's general applicability for measuring distribution shifts?
> While our primary motivation in this work is to develop a theoretically grounded distance measure that can accurately predict human perceptual judgments of image similarities, it is also natural to consider applying the same principles to other types of continuous signals, such as audio.
>
> We appreciate your question and have noted this in the last paragraph of our revised discussion section.
> Lastly, we would like to clarify that our current formulation and implementation are geared toward continuous data, but an interesting possibility is to extend them to handle discrete data such as text (as has recently been done with diffusion models).
>
> ## Question 2: In high-dimensional spaces (e.g., latent spaces), how can the "direction" of distributional change be effectively measured? How might the similarity or difference between these "directions" of change impact IEM's performance? Is IEM inherently sensitive to this directional information?
>
> The IEM is, by construction, sensitive to the direction of perturbation because its local metric $G(x)$ weights directions according to the local curvature of the log-density (see Eq. 5 and the theoretical analysis in Sec. 2, including Fig. 2). Consequently, two perturbations of equal Euclidean norm can produce very different IEM values, depending on their orientation relative to the level sets (iso-contours) of the log-density. As discussed in the second paragraph of the introduction, this anisotropic sensitivity is a widely noted property of human perception.
>
> To further illustrate this property, we have conducted additional experiments on image data (see the new Sec. 3.3 in the revised paper PDF). These experiments demonstrate that minimizing or maximizing the IEM under a fixed PSNR constraint yields substantially different outcomes, confirming empirically that the IEM exhibits direction-dependent, non-isotropic sensitivity.

---

> ### Author Response · Authors · 2025-11-20
> **Cont.**
>
> ## Question 3: Could IEM be applied to quantify the difference between generated and real images, thus serving as a metric for evaluating the performance of generative models?
> This is an excellent question. As discussed in App. B, the mismatched IEM (Def. 3) can be interpreted as a local positive-definite decomposition of the KL divergence between two distributions. While the mismatched IEM compares samples drawn from different distributions, it requires access to a denoiser for each of these distributions. Thus, it is not immediately applicable as a direct metric for evaluating the performance of a single generative model. However, it may be used to compare two generative models by identifying samples for which the mismatched IEM is largest—that is, where the models disagree most—analogous to recent work on discriminating between image representation models [1].
> We view this as a promising future direction for using the mismatched IEM, though this investigation lies beyond the scope of our current work.
>
> An alternative possibility is to use the IEM derived for real images to quantify similarity between corresponding pairs of real images and generated images – see response to reviewer o82F regarding "anomaly detection or out-of-distribution detection."
>
>
> [1] J Feather, D Lipshutz, S Harvey, A Williams, E Simoncelli. Discriminating image representations with principal distortions. In ICLR 2025.
>
> ## Question 4: How does IEM's performance compare directly to other information-theoretic measures like KLD, JSD, and EMD on your tasks? This comparison is vital for establishing IEM's value.
> We would like to clarify that the IEM provides a distance between individual samples, whereas the KL divergence (KLD), Jensen–Shannon divergence (JSD), and Earth Mover’s Distance (EMD) are statistical distances between entire distributions. However, as discussed in App. B, the IEM is mathematically connected to the KL divergence. Specifically, the KL divergence between two translated copies of the same density equals half the expected squared IEM. This relationship provides a precise mathematical bridge between the local geometric differences between the density and its translated version, as captured by the IEM, and the global difference between the density and its translated version, as captured by the KL divergence.
>
> ## Question 5: What explains the contradictory performance of $IEM_{sq.}$?
> You are correct to note this contrast (as mentioned in L445 in the original submission). The IEM performs strongly on LIVE, TID2013, and CSIQ, but less so on TQD, whereas $IEM_{sq.}$ performs well on TQD. We hypothesize that this occurs because TQD contains very different types of distortions compared to the other datasets (as discussed in L435 of the original submission). Specifically, TQD primarily contains large, "global" distortions (with substantial MSE), whereas the other databases contain various types of "local" distortions (with much smaller MSE) that are commonly encountered in image processing systems (e.g., noise, blur, and JPEG compression artifacts). As explained from L360 onward, the generalized $IEM_f$ (Def. 2) enables us to address different types of distortions, both local and global. The specific variant $IEM_{sq.}$ discounts higher-order differences across integration scales by weighting the integrand in Def. 2 using the squared value of the log-probability ratio (Eq. 9). If we assume the log-probability ratio (Eq. 9) for the two textures is small, the IEM will be less sensitive to global textural changes. However, this comes at the cost of discounting local distortions that may be perceptually noticeable. In contrast, the standard IEM (Def. 1) is more sensitive to global distortions such as texture variations, and therefore performs less effectively on TQD: it is better suited to datasets in which distortions are spatially localized or near the perceptual threshold, such as LIVE, TID2013 and CSIQ.
>
> Together with the learned variant $IEM_{f_{\omega}}$, where $f_{\omega}$ is a simple neural network trained in a semi-supervised manner while the denoiser remains frozen and entirely unsupervised, our results demonstrate that the IEM framework can flexibly adapt to different distortion regimes by modulating the integration weighting through the function $f$ in Def. 2. A compelling open question mentioned in the discussion is how to select $f$ systematically and in an unsupervised manner, so that the generalized IEM works for both local and global distortions.

---

> ### Author Response · Authors · 2025-11-20
> **Cont.**
>
> ## Question 6: The performance is highly sensitive to $\Gamma$. What is the underlying reason for this sensitivity?
> Thank you for this question. As noted in the discussion section, $\Gamma$ sets the upper bound of integration in Def. 1 and thus determines the finest scale at which the geometry of the data distribution is probed. In theory, taking $\Gamma\rightarrow\infty$ would capture all scales, but in practice, this is limited by the accuracy of the learned denoiser. Indeed, at very high SNR levels, where the denoiser must resolve fine-grained image details, its estimates of the score function become less reliable, and a large value of $\Gamma$ can introduce noise and instability into the metric. Conversely, a small value of $\Gamma$ truncates the integration, which makes the distance to ignore important fine-grained image structure (in fact, as $\Gamma$ approaches $0$, the IEM approaches the Euclidean distance).
> While this sensitivity is currently a limitation of the method, we view it as an interesting direction for future work. In particular, we hypothesize that $\Gamma$ could be chosen adaptively based on the performance of the denoiser across scales, for example, by monitoring the stability or variance of its denoising errors. Such an approach could potentially provide a principled, data-driven method for selecting $\Gamma$.
>
> ## Question 7:  Is the performance of the proposed method sensitive to the selection of the specific denoiser used?
> Thank you for this question. Similarly to diffusion models, the performance of the IEM indeed depends on the quality of the denoiser, which must provide an accurate estimation of the data distribution’s score function across different noise levels. For example, a linear denoiser would yield an IEM equivalent to the Mahalanobis distance, which would perform poorly on complex data such as natural images due to its lack of local adaptivity. For this reason, we employed in Sec. 3 a high-performing yet computationally efficient denoiser architecture to demonstrate our method. In principle, the IEM can be computed using any denoiser, and exploring larger or more specialized architectures may further improve performance.

---

> ### Comment · Reviewer_CAxo · 2025-11-24
> **Reply to the Response**
>
> I thank the authors for their response. However, regarding the points on additional experimental validation, scope of application, diverse denoisers, and comparative analysis, the replies remain largely subjective assertions. I believe objective experimental validation is necessary.
>
> Even for the IQA task, while the utilized IQA datasets contain diverse distortions, they primarily consist of single distortions. Different distortions (e.g., noise, blur) likely induce distinct directions of distribution shift. Given that the proposed method emphasizes the importance of this direction for measurement, it is crucial to evaluate its robustness under more realistic and challenging conditions involving mixed distortions. Therefore, I recommend including experiments on images with mixed degradations to substantiate the claims at least.

---

> > ### Author Response · Authors · 2025-11-25
> > **Reply to reviewer CAxo**
> >
> > We thank the reviewer for responding to our rebuttal.
> >
> > 1. __Regarding comparative analysis with KLD, JSD, and EMD:__ as we expressed in the paper and the rebuttal, the IEM is a distance measure between a pair of vectors in $\mathbb{R}^d$. Please allow us to clarify again that the IEM is not a distance between distributions. Thus, we are unsure how to conduct a meaningful comparison between the IEM (a distance between vectors) and KLD/JSD/EMD (a distance between distributions), as the reviewer suggested in the weakness "lack of comparative analysis." Could the reviewer please specify what kind of comparison they envision? An example from the literature where such a comparison has been carried out would be very helpful.
> > 2. __Regarding objective evaluation:__ as noted in the paper and the rebuttal, we have already evaluated the IEM on the five large-scale IQA databases, all of which are objective evaluations that constitute common practice in the field (BAPPS, TID2013, LIVE, CSIQ, TQD; see [1,2] for example, which are two widely known and highly cited papers). If the reviewer still considers our evaluations as insufficient, could they please specify which additional database(s) would make the evaluation “objective,” and explain why? We would also like to clarify that the new results we have added to the revised manuscript are not meant to replace the objective evaluations we have already conducted. These new results simply provide additional valuable insight into the behavior of the IEM.
> > 3. __Regarding mixed degradations:__ our current evaluation already encompasses a wide variety of mixed degradations. For example, BAPPS contains 308 sequentially composed distortions (see Sec. 2.1 and Tab. 1 in [2]), and TID2013 includes distortions produced by image denoising methods (a cascade of noise addition and removal), lossy compression of noise images (a cascade of noise and compression), as well as sparse sampling followed by reconstruction (see https://www.ponomarenko.info/tid2013.htm). If these widely used databases are still unsatisfactory, could the reviewer please explain why, and specify what kinds of mixed distortions (not already covered by the above) they consider “crucial?” Pointing to a publicly available IQA database that contains these mixed distortions would be helpful.
> >
> > [1] K Ding, K Ma, S Wang, and E Simoncelli. Deep image structure and texture similarity (DISTS) metric. In IEEE Transactions on Pattern Analysis and Machine Intelligence, 2022.
> >
> > [2] R Zhang, P Isola, A Efros, E Shechtman, and O Wang. The unreasonable effectiveness of deep features as a perceptual metric. In CVPR, 2018.

---

### Official Review · Reviewer_o82F · 2025-11-08

**Soundness:** 3
**Presentation:** 3
**Contribution:** 3
**Rating:** 8
**Confidence:** 4

**Summary:**

The authors introduce a novel distance metric called the Information-Estimation Metric (IEM). The IEM is defined based on fundamental principles connecting probability density with local geometry (i.e., connecting the estimation error to the gradient of the log of the marginal). Authors provide a rigorous theoretical framework, as well as experiments to further explain the usefulness of the proposed metric. Hence, I recommend this paper to be accepted.

**Strengths:**

The paper is well-written, the motivation is clear and sound. Theoretical framework is thorough, experimental section is provided. While the computational complexity can be an issue, authors still provided experimental results, which is appreciated.

The paper introduces a new distance metric that relies on the geometry of the distribution. The idea is interesting and useful. One strength is that the method is not informed by any downstream task, and is rather derived directly from the probability distribution of the data. Second, the IEM does not rely on the manifold hypothesis, and is well-defined for any valid probability density.

The IEM depends directly on the prior $p_{\mathbf{x}}$, rather than through an observation model, i.e., the representation $p_{\mathbf{y} | \mathbf{x}}$, that may or may not be dependent on the prior. And, the IEM is a proper metric, from which the authors derive a local metric $\mathbf{G}$ such that it is a one-way relationship. I.e., the IEM is not a geodesic distance.

Lastly, the experiments show interesting results -- the distance metric can predict human perceptual judgement.

Overall, the paper presents a significant theoretical advancement in connecting information theory, estimation theory, and geometry to derive perceptually meaningful distance metric.

**Weaknesses:**

An obvious weakness, as the authors acknowledge in the Discussion section, is the computational complexity of the approach. Numerical estimation of an integral is expensive, and there exist other works that provide a metric but have a smaller cost. It would be great if authors could come up with a way to make this practical and usable for the community. Also, the method needs training a diffusion model, which is also expensive.

The paper provides a metric for continuous distributions, which is often not available or well-defined for all types of data.

The experiments are focused on natural images. One of the strengths that the authors mention is that the IEM does not rely on the manifold hypothesis. While I understand that the theory of IEM does not rely on the manifold hypothesis, it strikes me that natural images \textit{do} lie on a manifold. It would make the paper stronger to see experiments on data that do not necessarily lie on a manifold (e.g., text).

Despite these limitations, the paper presents a significant theoretical advancement in connecting information theory, estimation theory, and geometry to derive perceptually meaningful distance metrics.

**Questions:**

How do authors expect to solve the computational complexity issue?

Have authors conducted experiments on other types of data?

The authors show that for Gaussian distributions, the IEM coincides with the Mahalanobis distance. Are there other known probability distributions where the IEM has a closed-form expression?

How does the IEM relate to other geometrically-informed distance measures like optimal transport distances (e.g., Wasserstein distance)?

The paper focuses on supervised versus unsupervised approaches. Could a semi-supervised approach yield better results with less labeled data?

How stable is the IEM to small perturbations in the underlying data distribution? Is there a way to quantify its robustness?

How might the IEM perform in domains where human perceptual ground truth is unavailable or difficult to obtain?

Could the IEM be applied to other perceptual tasks like anomaly detection or out-of-distribution detection?

How does the local metric G(x) derived from the IEM compare to other local metrics derived from different principles in manifold learning?

---

> ### Author Response · Authors · 2025-11-19
>
> We are grateful to the reviewer for their effort in reviewing our work and for providing insightful comments and questions. Our responses are provided below.
>
> ## 1. How do authors expect to solve the computational complexity issue?
> You are absolutely right that the IEM is more computationally demanding than standard perceptual metrics, which typically require a single network evaluation. This is because the IEM involves an integral over denoising errors, whose numerical computation requires multiple evaluations of a denoiser (analogous to the fact that diffusion models typically require multiple denoiser evaluations to generate samples). While we indeed mention this as a limitation in our paper, we also view it as an opportunity for future work—much like the early stages of diffusion models, which initially required hundreds of denoising steps but later became far more efficient through improved formulations. We are hopeful that the IEM will see similar progress. For instance, we are currently exploring a formulation that replaces numerical integration with a single-pass evaluation—much like single-step diffusion sampling methods.
>
> ## 2. Have authors conducted experiments on other types of data?
> While our primary motivation in this work is to develop a theoretically grounded distance measure that can accurately predict human perceptual judgments of image similarities, it is also natural to consider applying the same principles to other types of continuous signals, such as audio (for which one may also use a diffusion model to compute the IEM). We appreciate your question and have noted it in the last paragraph of our revised discussion section as a potential area for future work.
>
> ## 3. Besides the Gaussian distribution, are there other known probability distributions where the IEM has a closed-form expression?
> Thank you for this insightful question. Computing the IEM in closed form requires access to the analytical form of the optimal MMSE denoiser for the corresponding prior density, under additive white Gaussian noise corruption. For a few distributions—such as the Gaussian, separable Laplacian, and certain mixture models (e.g., GMMs)—the MMSE denoiser can be expressed in closed form or as a simple one-dimensional integral. However, even when the denoiser is known analytically, it still does not mean that there exists a closed-form solution for the IEM integral in Def. 1, because the denoising errors may depend on nonlinear functions of the input—such as the Gaussian CDF for the Laplace prior, or ratios of sums of Gaussian PDFs for mixture models. As a result, the IEM integral rarely admits a closed-form solution. Finding families of distributions for which this is possible thus remains an open theoretical question.
>
> ## 4. How does the IEM relate to other geometrically-informed distance measures like optimal transport distances?
> We appreciate this very important question. While we showed in App. B how the IEM relates to the KL divergence, it is indeed natural to ask how it relates to other distance measures between distributions, particularly optimal transport (OT) distances, which are defined in terms of an underlying pointwise cost function. We have not yet established such a connection between the IEM and OT, but we view this as one of the most intriguing directions in our future research. A related interesting open question is the following: *What is the meaning of an optimal transport distance defined using the IEM as its local cost function?* Would the resulting distance between distributions inherit any desirable geometric or statistical properties from the IEM? We do not yet know the answer, but exploring such a formulation—where an OT distance is induced by a cost derived from an underlying data density—is conceptually interesting.

---

> ### Author Response · Authors · 2025-11-19
> **Cont.**
>
> ## 5. The paper focuses on supervised vs. unsupervised approaches. Could a semi-supervised approach yield better results with less labeled data?
> A central aspect of the IEM is its unsupervised formulation, which requires training only an unconditional denoiser. However, the supervised variant of the IEM that we describe in Sec. 3 (and detailed in Appendix E.3) is, in fact, semi-supervised. Specifically, we learn a simple weighting function $f_{\omega}$​ that adaptively weights the different noise scales in the integral of the generalized IEM (Def. 2). This function is trained using both labeled data from the KADID-10k dataset and unlabeled data in the form of paired image patches extracted randomly from the DTD texture dataset (where each pair is taken from the same image). The use of such unlabeled data relies on the premise that random patches extracted from a given texture image are typically perceived as similar by humans [1].
>
> While this constitutes a basic form of semi-supervised learning, it indeed raises an interesting question: whether combining limited labeled data with even more unlabeled data could achieve comparable or even superior alignment with human perception. Beyond the texture case we considered (where spatial proximity provides a natural unsupervised training signal), it remains unclear how to design unsupervised objectives that meaningfully reflect perceptual similarity in more general settings.
>
> [1] K Ding, K Ma, S Wang, and E Simoncelli. Image Quality Assessment: Unifying Structure and Texture Similarity. IEEE Transactions on Pattern Analysis and Machine Intelligence, 2022.
>
> ## 6. How stable is the IEM to small perturbations in the underlying data distribution? Is there a way to quantify its robustness?
> This is an important question. The answer depends on how one defines a perturbation of the data distribution. For instance, if the distribution is perturbed by adding small Gaussian noise to the samples, then the resulting change in the IEM is negligible, since the integration in Def. 1 is computed over a range of noise levels (from 0 to $\Gamma$). More generally, analyzing the sensitivity of the IEM to other forms of perturbations of the underlying prior can be approached in at least two ways:
> 1. By examining how the local metric $G(x)$ changes under transformations of the density (e.g., shifts, scalings, or smooth deformations).
> 2. By evaluating the mismatched IEM (App. B) between a point and itself under the two different priors, and aggregating these quantities across the domain. A large mismatched IEM between two closely related distributions would indicate an issue with the trained denoiser.
>
> Both approaches could provide a principled way to quantify the robustness of the IEM to small perturbations of the distribution. However, this is beyond the scope of our paper, and we leave it as an interesting direction for future work.
>
> ## 7. How might the IEM perform in domains where human perceptual ground truth is unavailable or difficult to obtain?
> We would like to emphasize that the IEM is particularly appealing in such domains, where human perceptual ground truth is unavailable or difficult to obtain, precisely because the IEM is entirely unsupervised: it is defined directly in terms of unlabeled data through a learned denoiser, and does not require human labels. Notably, while the generalized IEM can incorporate supervision (e.g., through the learned weighting function $f_{\omega}$, as shown in Sec. 3), we view this as an advantage of the approach: it is unsupervised by design, yet it can be further enhanced when supervised data are available.
>
> ## 8. Could the IEM be applied to other perceptual tasks like anomaly detection or out-of-distribution detection?
> Thank you for this interesting question—this is indeed one of the main directions we are currently exploring. We believe that the IEM can be adapted for anomaly and out-of-distribution (OOD) detection. For example, one could measure the average IEM between a given sample $x_0$ and a random sample from the distribution, via $\mathbb{E}[\text{IEM}(X, x_0)]$, where the expectation is taken with respect to $p_X$​. For a Gaussian prior, this reduces to the Mahalanobis distance between the point $x_0$ and the mean of the distribution—a technique widely used for anomaly detection in low-dimensional spaces.
> However, in high-dimensional spaces, this approach in its basic form is likely to fail. Indeed, even for a zero-mean Gaussian distribution, the zero vector ($x_0=0$) minimizes the Mahalanobis distance to the mean, yet the zero vector is not a typical sample from such a distribution (due to the concentration of measure phenomenon), and is thus generally considered as an outlier. We are currently investigating how the IEM, or a refined notion of anomaly derived from it, could better account for these high-dimensional effects and serve as a principled framework for anomaly and OOD detection.

---

> ### Author Response · Authors · 2025-11-19
> **Cont.**
>
> ## 9. How does the local metric G(x) derived from the IEM compare to other local metrics derived from different principles in manifold learning?
> As discussed in App. A, the local metric $G(x)$ derived from the IEM is fundamentally different from manifold-learning-based local metrics. In particular, manifold learning methods typically assume that data lie on a low-dimensional manifold, and the local metric is estimated from pairwise distances between samples—that is, a predefined distance function is required to infer the local geometry. In contrast, the IEM and its local metric $G(x)$ are derived directly from the data density through a denoising-based formulation, and are therefore well-defined even when the data support is not manifold-like.
>
> Moreover, traditional manifold learning techniques are generally limited to analyzing a finite set of samples, and extending the learned metric to unseen data is not straightforward. In contrast, the IEM and the corresponding local metric $G(x)$ are defined continuously over the entire data domain, enabling distance computation between any pair of points and local metric evaluation at any location, including new test datapoints that were not available during training.
>
> Lastly, as you noted, the local metric $G(x)$ is derived from the global IEM distance. However, as discussed in Sec. 2, this relationship is one-way: in general, the IEM is not the geodesic distance induced by $G(x)$. In contrast, manifold learning methods typically proceed in the opposite direction—they start from a local metric and define the global distance as the geodesic obtained by integrating that local metric.
>
> ## 10. A note about the manifold hypothesis
> We appreciate the reviewer's point about the manifold hypothesis. While it is commonly believed that natural images lie on or near a low-dimensional manifold within the high-dimensional pixel space, it is worth noting that recent evidence challenges this hypothesis. In particular, the work in [1] shows that the “local dimensionality of the energy (log-probability) landscape in the neighborhood of an image varies greatly depending on image content and neighborhood size.” Moreover, they find “both images with full-dimensional neighborhoods of non-negligible size, and images with lower-dimensional neighborhoods even at sub-quantization scales.” Therefore, while natural images are highly structured and exhibit strong statistical dependencies, they are not necessarily confined to a low-dimensional manifold (at least not in the mathematical sense, as manifolds have a fixed dimensionality).
>
> As you noted, the IEM does not make any such manifold assumption: it can flexibly adapt to the varying shape and dimensionality of the data support.
>
> [1] F Guth, Z Kadkhodaie, and E Simoncelli. Learning normalized image densities via dual score matching. In Advances in Neural Information Processing Systems (NeurIPS), 2025.

---

> > ### Comment · Reviewer_o82F · 2025-11-26
> > **Thank You for Your Answer**
> >
> > I thank the authors for their detailed and thoughtful responses. I will keep my score as is.

---

### Official Review · Reviewer_hJYw · 2025-11-08

**Soundness:** 4
**Presentation:** 4
**Contribution:** 4
**Rating:** 8
**Confidence:** 3

**Summary:**

The paper introduces a well-defined (proper) distance metric called the Information-Estimation Metric (IEM) induced by a probability distribution. Specifically, the metric is computed by comparing the denoising errors between the input signals when subject to varying levels of noise. ​It is built on the pointwise ​I-MMSE and the Tweedie-Miyasawa formulas. Inspiration is also drawn from the generative process in a diffusion model. The metric is shown to be the Mahalanobis metric when the underlying distribution is Gaussian. As an application, it is shown to correlate well with human judgment when applied to image pairs. Other interesting applications are identified for future investigation.

The paper makes a solid contribution with several potential follow-up avenues. The experiments are both illustrative and convincing, and the writing is top-notch. ​

**Strengths:**

​The formulation is novel and has many potential applications.

​The exposition and the illustrations are excellent.

**Weaknesses:**

A better connection between IEM and the image quality experiment could be made.

**Questions:**

Could you please provide some intuition on how iso-IEM contours look like at larger scales (or for larger epsilon)?

As a follow-up question, can you please comment on the IEM landscape in the context of image quality assessment? Specifically, how do images that lie on an iso-IEM contour or a ball with the reference at the center look like? How do they compare with, say, iso-VIF or iso-LPIPS images?

Can you please point to the prior distribution assumed for the IQA experiments? I may have missed this detail.

Why does the IEM correlate well with human judgment?

Can you please comment on the motivation for exploring the connection with the KL divergence? From what I understand, the IEM is defined for points drawn from the same underlying prior data distribution. The KL divergence, on the other hand, compares distributions. The translation of p_x does answer this to some extent, but some more insights could help the readers.

---

> ### Author Response · Authors · 2025-11-19
>
> We sincerely thank the reviewer for taking the time to evaluate our work and for providing very insightful comments and questions.
> We address the reviewer's questions below.
>
> ## 1. Could you please provide some intuition on how iso-IEM contours look like at larger scales (or for larger epsilon)?
> Certainly. The IEM between two images $x_1$ and $x_2$ captures the difference in the local geometry (level sets) of the log-density in their respective neighborhoods. Following the definition of the IEM, each neighborhood can be viewed as being weighted by an effective window that is aggregated across scales, centered at either $x_{1}$ or $x_{2}$. These induce two windowed versions of the density, which are then compared in a way that resembles a KL divergence.
>
> This interpretation can help build intuition about the shape of the iso-IEM contours at larger scales. Specifically, one can imagine comparing the windowed level sets of the log-density across different regions of the domain. For example, in the top-right panel of Fig. 2, which depicts equidistant-IEM contours for a Gaussian mixture prior, we observe non-monotonic equidistant contours relative to the indicated reference point $x_{\text{ref.}}$. Moving upward and left from the reference point initially increases the IEM, but as one approaches the second mode of the distribution, the local level sets of the log-density become more similar to those around the reference point, and the IEM decreases.
>
> We hope this explanation helps clarify the intuition behind the IEM’s behavior. We would be happy to provide additional details or illustrations if you believe they would be helpful.
>
> ## 2. How do images that lie on an iso-IEM contour or a ball with the reference at the center look like? How do they compare with, say, iso-VIF or iso-LPIPS images?
>
> Thank you for raising this important question, which prompted us to conduct additional experiments that yielded new and meaningful findings. While it is not immediately clear how we could optimize natural images to attain a specific IEM value (in order to visualize the images that lie on an iso-IEM contour around a reference image), we can instead minimize or maximize the IEM (or other metrics) while keeping the PSNR fixed, effectively conducting a maximum differentiation competition [1] against PSNR. This corresponds to finding, according to a given perceptual metric, the best or worst images lying on a hypersphere of fixed radius in $\mathbb{R}^d$, as illustrated in Fig. 5 that we’ve added to the revised paper PDF.
>
> Implementing this procedure requires optimizing the IEM, and we have identified a practical strategy to achieve this. In particular, our solution involves annealing the noise level of the integrand in the definition of the IEM throughout the optimization—see our new Sec. 3.3 and App. D.3 in the revised paper PDF for more details. We conducted such an experiment and compared the results with those of other metrics. We found that optimizing the IEM yields high-perceptual-quality images that preserve the overall geometric structure of the reference image, even under substantial deviations of  PSNR=10dB. In contrast, optimizing each of eight other metrics (SSIM, VIF, LPIPS, DISTS, TOPIQ, PieAPP, NLPD, GMSD) yields more objectionable artifacts (as illustrated in Figs. 5 and 9-13 in the revised manuscript).
>
> This is an encouraging (and nontrivial) result: while a distance function may align well with human perceptual judgments on a limited set of distortions (e.g., those that have been tested on humans in a particular experiment), this does not imply that such a distance function can serve as a robust optimization objective, since it may fail to respond correctly to other types of distortions. Our new experiments suggest that the IEM may indeed serve as a robust objective.
>
> [1] Z Wang and E Simoncelli. Maximum differentiation (MAD) competition: A methodology for comparing computational models of perceptual quantities. Journal of Vision, 2008.

---

> > ### Author Response · Authors · 2025-11-19
> > **Cont.**
> >
> > ## 3. Can you please point to the prior distribution assumed for the IQA experiments? I may have missed this detail.
> > The IEM is defined in terms of a probability density over images, which in our experiments is the one that is implicitly embedded in a denoiser trained on the ImageNet dataset, without using class conditioning (see details in L349 and Appendix E.2). That is, we are using a diffusion model for the prior, which captures the statistical properties of the training data (in the case of ImageNet, a broad set of natural scenes to which humans are typically exposed).
> >
> > ## 4. Why does the IEM correlate well with human judgment?
> > While the precise mechanisms underlying human perception remain only partially understood, we can provide a hypothesis for the strong correlation between the IEM and human judgments.
> >
> > First, theories of efficient coding posit that early visual processing performs a form of “whitening” of natural signals—decorrelating their components to reduce redundancy and maximize information transmission, under biological constraints [1]. The IEM embodies a similar principle: locally, it behaves as a Mahalanobis metric, which is equivalent to performing local whitening of the signal. In other words, the IEM adapts to the local covariance structure of natural images, consistent with the transformations hypothesized in early biological sensory processing.
> >
> > Second, when the underlying data distribution is heavy-tailed, as is typical for natural images [2,3], the IEM becomes more sensitive in high-probability regions where the local curvature of the log-density is larger (as discussed in our paper, e.g., in Sec. 2.2). This property aligns with predictions of efficient coding and psychophysical evidence, which show that humans are more sensitive to perturbations in images they see more frequently [4,5].
> >
> > [1] B Olshausen and D Field. Emergence of simple-cell receptive field properties by learning a sparse code for natural images. Nature, 1996.
> >
> > [2] D Field. Relations between the statistics of natural images and the response properties of cortical cells. J. Opt. Soc. Am. A., 1987.
> >
> > [3] J Daugman. Entropy reduction and decorrelation in visual coding by oriented neural receptive fields. IEEE Trans. Biomed. Eng., 1989.
> >
> > [4] D Ganguli and E Simoncelli. Efficient sensory encoding and Bayesian inference with heterogeneous neural populations. Neural Computation, 2014.
> >
> > [5] X Wei and A Stocker. Lawful relation between perceptual bias and discriminability. Proceedings of the National Academy of Sciences, 2017.
> >
> > ## 5. Can you please comment on the motivation for exploring the connection with the KL divergence?
> > This is an important point that we failed to communicate clearly. Since our work introduces a new mathematical object, the IEM, it is useful to study its properties in order to gain a better understanding of it. One way to do so is to understand the relationship between the IEM and previously established and well-studied quantities. The relationship we establish between the IEM and the KL divergence enables us to interpret the IEM as a local decomposition of the KL divergence between a distribution and a  translated copy of the same distribution. This is analogous to the interpretation of the Mahalanobis distance as a KL divergence between two translated copies of the same Gaussian distribution:
> > \begin{align}
> > D_{KL}(\mathcal{N}(\mu+x_1,\Sigma)|| \mathcal{N}(\mu+x_2,\Sigma)) = \frac{1}{2}(x_1-x_2)^{\top} \Sigma^{-1}(x_1-x_2)
> > \end{align}
> > Moreover, this connection opens the door to potential generalizations. In particular, one could construct analogous metrics based on other $f$-divergences by decomposing them through modified forms of the pointwise I–MMSE relation. This could yield a broader family of information-estimation-geometry-based distance functions between points, potentially offering variants that are easier to compute or that emphasize different geometric properties of the density.

---

> > ### Comment · Reviewer_hJYw · 2025-11-26
> > **Response to rebuttal**
> >
> > The answers provide the insights I was looking for and would certainly help readers in better understanding the contributions of this work. The answer to the second question is especially interesting and intriguing. The authors' efforts in conducting the additional experiments and revising the illustrations/paper are appreciated. The theoretical contribution is complemented well by the IQA application. Overall, this is a solid paper. I retain my original rating.

---

### Author Response · Authors · 2025-11-20
**Revision summary**

We thank the reviewers for their efforts in evaluating our paper and providing valuable feedback. In response to the reviews, we have further improved our work by conducting additional experiments in Sec. 3.3, with new visual illustrations in Figs. 5 and 9–13, and further details in App. D.3.

In particular, Sec. 3.3 (and App. D.3) introduces a new optimization procedure for finding images that are closest to (or furthest from) a reference image according to the IEM, while remaining on a hypersphere in $\mathbb{R}^{d}$ centered on the reference image (i.e., with a fixed MSE). Interestingly, we find that minimizing the IEM yields artifact-free images that are perceptually similar to the reference image, whereas minimizing state-of-the-art perceptual metrics yields unrealistic images with noticeable artifacts. This demonstrates that the IEM generalizes well over the full space of images: not only does it detect the specific distortions that have been tested on humans in a specific experiment (e.g., those in the LIVE, TID2013, CSIQ and TQD databases), but it is also able to anticipate and handle perceptually noticeable distortions that may arise during optimization.

We hope this summary of our new experiments will streamline your review process and facilitate your evaluation of the updates.

---

### Meta-Review · Area_Chair_ntVU · 2026-01-05

**Summary:**

Reviewer hJYw notes that the connection between the proposed IEM and the image-quality experiments could be strengthened.
Reviewer o82F raises concerns about the high computational cost of the method and the limited scope of experiments, which focus mainly on natural images.
Reviewer CAxo also points out limited modality coverage and suggests broader validation beyond visual data.
Reviewer dR9Y emphasizes the relatively low computational efficiency and the restricted range of noise types evaluated.

These concerns were weighed against the strong theoretical contribution, clear exposition, and solid experimental validation in recommending acceptance.

**Reviewer Concerns:**

Reviewers hJYw and o82F’s concerns have been fully addressed, and Reviewer dR9Y is generally satisfied with the authors’ response. Reviewer CAxo still raises some concerns regarding experimental validation, scope of application, the use of diverse denoisers, and comparative analysis. However, after reviewing the authors’ response, I believe these concerns have either been addressed or are not sufficiently specific to be meaningfully addressed by rebuttal.

**Reviewer Scores:**

Reviewer hJYw: 8
Reviewer o82F: 8
Reviewer CAxo: 6
Reviewer dR9Y: 6

Since the original ratings are generally positive and the reviewers’ concerns have been largely addressed in the rebuttal, I expect that they will maintain their scores.

---

### Decision · Program_Chairs · 2026-01-26

Accept (Poster)